# Leveraging Knowledge Graphs to harvest a high-quality dataset for efficient CLIP model training

## Abstract

Vision-language contrastive learning based on the CLIP method has been instrumental in driving recent advancements in computer vision. However, high quality CLIP models are based on very large datasets. This makes them expensive to train and hampers the scientific analysis of these models. We show how to train a CLIP base-size model efficiently for a broad domain on a much smaller amount of data. We demonstrate this specifically with the automated creation of a dataset named LivingThings with 8.9M images of animals and plants and 12.2M texts. The dataset is obtained via focused image-search queries of three kinds: entity queries (e.g., "eagle"), entity-attribute queries (e.g., "bushy tail of a fox"), and type-attribute queries (e.g., "insect on a leaf"). The entities and types, as well as some of the texts, are derived from the WordNet and Wikidata knowledge graphs, the attributes are obtained via LLMs. We train a CLIP model from scratch on LivingThings and evaluate it on ImageNet, iNaturalist, and CUB for object classification and OVAD and CUB for attribute classification. On the broad target domain of animals and plants, our model achieves comparable, and sometimes even much better performance than models that have orders of magnitude more parameters or training data. For instance, our ViT-B-32 model improves over much larger state-of-the-art CLIP models on the iNaturalist 21 object classification task. We will publicly release our code and dataset.

## 1 Introduction

Contrastive Language-Image Pretraining (CLIP) (Radford et al., 2021) is a popular way to train Vision-Language Models (VLMs) and is part of a large percentage of contemporary works in computer vision. CLIP models learn high-quality visual embeddings and establish a link to the semantic level of brief text descriptions by training on pairs of images and their corresponding text descriptions collected from the web. The features and the link between images and text have been used directly for, e.g., zero-shot classification or text-to-image retrieval, and enable dialogues with visual input, such as in the LLaVA family of models (Liu et al., 2023). The link can also be exploited in the opposite direction to enable text-conditional image generation, e.g., Stable Diffusion (Podell et al., 2023).

However, pretraining such a model is very expensive, since it requires large amounts of data and compute, with the original CLIP model already training on 400M image-text pairs, and later works scaling the training up even more (Gadre et al., 2023; Fang et al., 2024a). Therefore, research on VLMs that requires control over the training of the model is either left to companies or is limited to models of significantly lower quality. Finetuning existing CLIP models can be done with moderate compute, however, then the architecture and pretraining data is fixed. Pretraining allows us to be in full control of all input data, choice of architecture and training algorithm.

The goal of this work is to train a VLM from scratch with much less compute while approaching or even surpassing the quality of the largest models. Our strategy focuses on training with less but better data, in order to maximize the performance per datapoint. Li et al. (2024) have explored CLIP "along three dimensions: data, architecture, and training strategies" and they stress the "significance of high-quality training data". For LLMs, Abdin et al. (2024) have shown that data curation brings

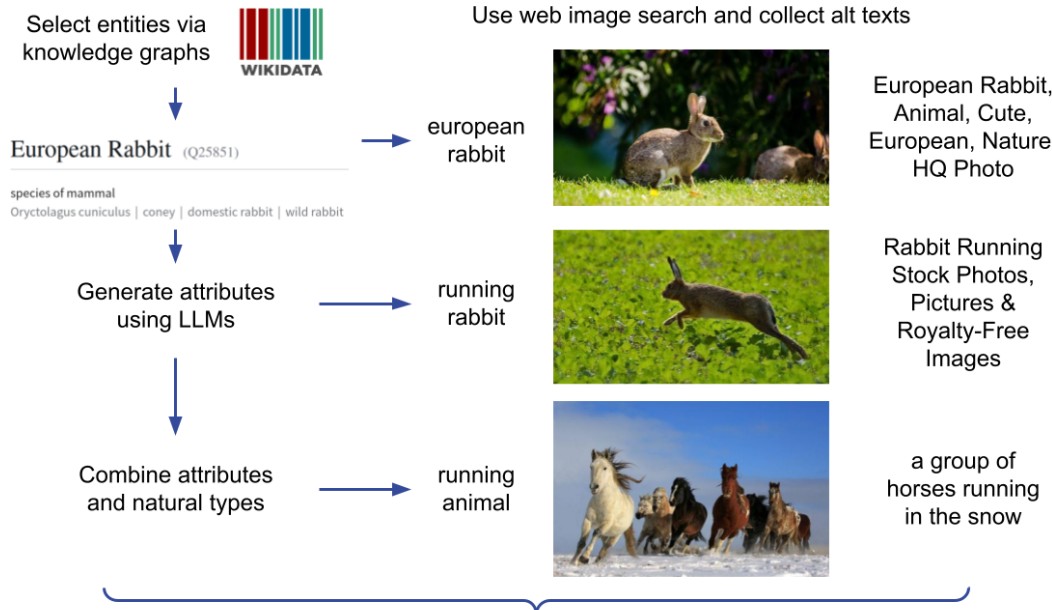

Figure 1: We create a dataset for vision-language pretraining: First, we extract entities from knowledge graphs, then generate attributes and natural types for them. We search for different combinations of entities, attributes, and types in image search engines, and collect alt texts for each image. We train our model on the combined data.

down the training time and model size, which they achieve via "heavily filtered publicly available web data and synthetic data".

In this paper, we try to achieve this goal in a largely automated fashion by leveraging the very compact and high-quality information found in knowledge graphs, specifically, we make use of WordNet (Fellbaum, 1998) and the bigger and more comprehensive Wikidata (Vrandečić & Krötzsch, 2014) to build a dataset for the domain of animals and plants. This domain is broad enough to be useful and serve as a proof of concept, and allows us to evaluate the performance of our model in detail for each domain and on existing benchmarks. We still obtain a foundation model in the sense that the model — within the semantic domain of animals and plants — can be used in an open-vocabulary, zero-shot manner. Furthermore, our dataset-building method is generic, and can be used for arbitrary domains covered by the given knowledge graphs.

We consider the following as our main contributions:

- We demonstrate how to build an effective dataset for a rather broad domain that enables training CLIP-like models from scratch on small-scale hardware. The dataset is obtained via focused image-search queries of three kinds: entity queries (e.g., "eagle"), entity-attribute queries (e.g., "bushy tail of a fox") and type-attribute queries (e.g, "insect on leaf"). The entities and natural types are obtained from the WordNet and Wikidata knowledge graphs, the attributes are obtained via LLMs. We also use the knowledge graphs to generate additional text labels for the retrieved images. The resulting LivingThings dataset comprises 8.9M images of animals and plants and 12.2M texts. The generation method of our dataset is largely generic and can be applied to arbitrary domains.

- We train a CLIP model on LivingThings from scratch, using low amounts of compute. We evaluate our model and a suite of other CLIP models on ImageNet, iNaturalist, Caltech-UCSD Birds (CUB), and RareSpecies for object classification, and on OVAD and CUB for attribute classification. Specifically for CUB, we create a comprehensive evaluation setup to enable testing zero-shot attribute classification using VLMs. On the target domain of animals and plants, our model achieves comparable and sometimes much better performance than models with orders of magnitude more parameters or training data.

## 2 RELATED WORK

**Datasets.** Many recent works investigate ways of building large-scale datasets for multimodal training. Radford et al. (2021) train the original CLIP model on a closed set of 400M images, with the model weights being released, but not the data. They build their dataset by collecting image-text pairs, where the text includes frequent terms derived from Wikipedia or WordNet nouns, and approximately class-balancing the result. As a first approach to create a public dataset of this size, Schuhmann et al. (2021) build a dataset with 400 million image-text pairs by filtering HTML data from Common Crawl (LAION-400M) (Rana, 2010). Their main method of filtering is to remove all image-text pairs that have less than 0.3 similarity estimated by the CLIP model. In a follow-up work, Schuhmann et al. (2022) scale their approach up one order of magnitude with the multilingual LAION-5B dataset. Xu et al. (2024) intend to replicate the original CLIP's data curation approach. They collect image-text pairs from CommonCrawl and filter them using Wikipedia and WordNet, then balance the results. Gadre et al. (2023) propose DataComp, a filtering challenge with a candidate pool of up to 13B image-text pairs from CommonCrawl, where the goal is to filter this candidate pool and run a fixed training pipeline on the resulting data. They propose a baseline DataComp-1B dataset with 1.4B pairs filtered with a combination of CLIP score and clustering CLIP embeddings to find images close to ImageNet (Deng et al., 2009) training examples. Fang et al. (2024a) train a Data Filtering Network on an internal dataset of 357M human-verified image-text pairs and finetune it on a set of public human annotated datasets. They filter 42B candidates into the DFN-5B dataset and train the current top model of the OpenCLIP leaderboard (Ilharco et al., 2021).

These large datasets have mostly replaced smaller datasets like ConceptualCaptions12M (CC12M) (Changpinyo et al., 2021), which relies on unimodal heuristics as well as Google Cloud Vision APIs to predict the image-text similarity. Another popular small dataset is Yahoo Flickr Creative Commons 15M (YFCC15M), a subset of 15M image-text pairs obtained from the YFCC100M dataset which is based on Flickr (Thomee et al., 2016). Though many works are mostly concerned on scaling up multimodal datasets and models as much as possible, we aim to improve research on high-quality CLIP models also in scenarios with much less available compute.

Stevens et al. (2024) aim to create a general vision model for organismal biology and curate the TreeOfLife-10M dataset based on Encyclopedia of Life (EOL, 2018), iNaturalist 2021 (Van Horn et al., 2018) and BIOSCAN-1M (Gharaee et al., 2023). Their model BioCLIP is trained on a mix of english and latin entity names. For evaluation, they curate the RareSpecies benchmark which tests generalization to 400 species unseen during training. While Stevens et al. (2024) use biological domain knowledge to build their dataset, we instead rely on knowledge graphs and propose a dataset collection method for arbitrary domains.

**Training algorithms.** Various works are concerned with improving CLIP from the algorithmic side. Li et al. (2023a) simply train on low resolution first, then finetune on higher resolution later. Li et al. (2023b) additionally mask a substantial portion of the image to further reduce the amount of input during training. Zhai et al. (2023) propose using a sigmoid loss which reduces the computional load especially in big distributed settings. Vasu et al. (2024) enhance their training data with synthetic captions created by an image captioning model and use an ensemble of CLIP teachers to train their model. This way, they can increase the learning efficiency by transferring knowledge from bigger models to their smaller models. Such algorithmic improvements are orthogonal to our research, since they would potentially also improve training on our dataset. In this work, we focus on data improvements and fix the algorithm and architecture choices, since this also allows to easily and fairly compare to a big set of already trained vanilla Vision Transformer (ViT) based CLIP models.

Li et al. (2024) scale down CLIP and analyse the influence of different data, architecture, and training strategies. They find that especially large models need larger datasets, and data quality plays an important role. They create higher quality datasets by applying CLIP filtering to the 3.4B WebLI dataset (Chen et al., 2023), while we aim to use a different dataset collection process.

**Evaluation.** Image classification is a popular way to evaluate vision encoders like CLIP-style models. While established benchmarks like ImageNet (Deng et al., 2009) or iNaturalist (Van Horn et al., 2018) provide a solid grounding to evaluate CLIP-style models on object classification, evaluating attribute understanding is more challenging. Attributes are more challenging to define and annotate: It is quite obvious what a "dog" is, however calling an object "large" depends on the frame of reference. The Animals with Attributes dataset (Xian et al., 2019) for example has attributes

annotated per class, which makes it unfavorable for testing per-image attribute classification. Bravo et al. (2023) find weaknesses in attribute definition and annotation of existing benchmarks like VAW (Pham et al., 2021) and provide the Open Vocabulary Attribute Detection (OVAD) benchmark, which can be used for evaluating both attribute classification and attribute detection. The CUB dataset proposed by Wah et al. (2011) provides images of 200 bird species, with 312 attributes densely annotated for each image. Since we propose a dataset built on entities and attributes derived from the world of animals and plants, the CUB dataset is an obvious choice to test our model.

## 3 DATASET CREATION

We describe our dataset creation process, consisting of four main steps: entity extraction, attribute generation, query building, and image search. In this work, our starting point for this process are entities within our target domain of animals and plants. In general, this process is applicable to all visual domains covered by the knowledge graph.

### 3.1 ENTITY EXTRACTION

We build our list of entities for both animal and plants from the Wikidata knowledge graph (Vrandečić & Krötzsch, 2014). We define an animal to be every entity that is a subclass or child taxon of the animal entity[1], either directly or via multiple intermediate entities. We exclude human individuals, mythical creatures, and other named individuals to avoid being too specific.

We define a plant analogously, but with the plant entity[2] as root entity instead. For plants, we also include their fruits, because they are typically not directly related to the plant entity via the taxon or subclass hierarchy. We exclude cultivars, named individuals and all plants with a coordinate location to avoid overly specific entities.

For every animal and plant, we download its identifier, name, description, number of Wikipedia sitelinks, aliases, taxon common names, and taxon names. See Tab. 1 for example data. We use the number of Wikipedia sitelinks as proxy for an entity's popularity and order our final entity lists by it, with popular entities coming first. Our final entity lists contain 204,918 animals and 84,612 plants.

We also select all nouns from WordNet (Fellbaum, 1998) that are a subclass of the "living thing" node, excluding humans, named entities and entities that cannot be seen with the bare eye, e.g., microorganisms. Finally, we only consider leaf nodes and arrive at 6,983 entities, each with a description and a total of 16,705 synonyms.

To verify that our entity extraction procedure generalizes beyond animals and plants, we run it on a much broader visual domain in Appendix I.

### 3.2 ATTRIBUTE GENERATION

We generate attributes for the top 500 Wikidata entities in both our animal and plant entity lists using LLMs. We manually define 6 visual attribute categories for plants and 7 for animals, for each of which we guide the LLM to predict between 1 and 10 attribute instances using constrained decoding. We thereby obtain between 6 and 60 attributes per animal and between 7 and 70 attributes per plant. The 6 attribute categories for plants are *Color*, *Pattern and texture*, *Plant parts*, *Shape and size*, *Habitat and environment*, and *Other*. For animals, we switch *Plant parts* to *Body parts* and add another category called *Behavior and movement*. We specifically prompt the LLMs to generate visually observable attribute instances.

We independently generate attributes for both animals and plants using four different open source LLMs [3] and merge their results afterwards. The LLMs generate around 18.6 attributes per animal or 19.7 per plant on average. After merging and deduplication, we end up with 44.4 attributes per animal and 56.7 attributes per plant. Note that for each attribute within an attribute category we also

---

[1]https://www.wikidata.org/entity/Q729
[2]https://www.wikidata.org/entity/Q756
[3]Mixtral 8x22B (Jiang et al., 2024), Mistral 7B (Jiang et al., 2023), Llama3 70B, and Llama3 8B (Meta, 2024). We use the non-instruct version for each of them, which we found to perform better than the instruct versions, especially when prompted with examples of the task.

Table 1: Examples of plant and animal entities and accompanying additional information as extracted from the Wikidata knowledge graph. The concrete graph queries can be found in Appendix E. Name, description and aliases are used as text labels during training. The number of sitelinks are considered a proxy for an entity's popularity. The name and aliases are used during search.

| Identifier | Name | Description | Sitelinks | Aliases / Common names / Taxon names |
| --- | --- | --- | --- | --- |
| Q5113 | bird | class of vertebrates characterized by wings, a feather-covered body and a beak | 264 | avian species / birds / Aves |
| Q19939 | tiger | species of big cat | 216 | tigress, tigers / tiger / Panthera tigris |
| Q11575 | maize | species of grass cultivated as a food crop | 216 | maize plant, corn, corn plant / Indian Corn, Teosinte / Zea mays |

Table 2: Examples of animal and plant attributes for different Wikidata entities and categories, generated by LLMs. Note that we also generate a search queries for all entity-attribute-combinations, which are later used for image search and during training.

| Wikidata entity | Attribute category | Attribute | Search query |
| --- | --- | --- | --- |
| dog | Pattern and texture | smooth | smooth dog fur |
| wolf | Habitat and environment | snow | wolf in the snow |
| buzzard | Body parts | talon | buzzard talons |
| garlic | Shape and size | big | big garlic bulb |
| rose | Other | dry | dried rose |
| cherry | Color | black | black cherry |

use the LLM to generate an appropriate search query containing the attribute and entity itself. This search query can then later be directly plugged into an image search engine. See Tab. 2 for examples of generated attributes and search queries.

## 3.3 QUERY BUILDING

Before building the search queries for our entities and attributes, we generate our entities' natural types. The natural type of an entity is the superclass that a human would most likely associate with it, e.g., *bird* for *eagle*, or *tree* for *oak*. It is neither too general nor too specific, and can be used to disambiguate the entity from other entities with the same name. We generate natural types for the first 5000 animals and plants in our entity lists with an LLM, and do this by asking the LLM to select a natural type entity from the entity's parent hierarchy. To all other entities we assign the natural type *animal* or *plant*. Incorporating natural types in the search queries improves the search results by reducing ambiguities for entities with names that carry multiple meanings, and by reducing the number of cartoons, illustrations, and other unwanted search results. For example, searching for *dove* returns mostly pictures of the well known personal care brand, whereas searching for *dove bird* returns pictures of the animal that we are actually interested in. We end up with 6 unique natural types for animals: bird, mammal, insect, fish, reptilia and animal itself. We also get 6 unique natural types for plants: tree, fruit tree, root vegetable, flowering plant, herb and plant itself.

We then generate three different types of search queries: Entity queries, entity-attribute queries and type-attribute queries. As entity queries, we combine the name of the entity and its natural type. For entity-attribute queries, we directly use the search queries generated by the LLMs (see Sec. 3.2). For type-attributes queries we search for all unique combinations of an entity's natural type with the attributes the LLMs generated for it. For example, we search for *flying bird* if for at least one entity whose natural type is *bird* the LLMs generated the attribute *flying*.

Table 3: Details of our LivingThings dataset. We show the number of unique elements for each column, e.g. the number of images after deduplication or all unique attributes create by LLMs after merging. Queries are split approximately evenly between the animal and plant domain.

| Query set | Images | Queries | Entities | Attributes | Alt texts | Example query |
|---|---|---|---|---|---|---|
| WordNet entity | 2,331k | 17k | 7k | - | 3,676k | kohlrabi |
| Wikidata entity | 4,372k | 56k | 56k | - | 5,604k | eurasian lynx |
| Wikidata entity + attribute | 2,714k | 47k | 1k | 5k | 4,408k | mature bald eagle |
| Wikidata type + attribute | 968k | 7k | 12 | 5k | 1,863k | tropic plant |
| All | 8,889k | 125k | 63k | 5k | 11,760k | - |

## 3.4 IMAGE SEARCH AND FILTERING

For all query sets, we search using both the Google Custom Search API and the Bing image search API. The Wikidata animal and plant entity query sets are very large, so we limit them to the top 28k entities each. For both entity-attribute queries and type-attribute queries, we use the full query sets with 47k and 7k queries, respectively. We activate the *SafeSearch* filters of the search engines.

Both search APIs also return the URL for the website an image is embedded in. We use that to download the corresponding HTML and search for the image tag matching the returned image in it. We then extract and store texts from attributes of the image tag as alt texts to use them later for training.

We thereby collect 4.7M / 15.6M images and 10.1M / 27.6M alt texts from the Google and Bing API, respectively. After downloading search results and alt texts, we postprocess the images and alt texts in the following way:

- Similar to Changpinyo et al. (2021), we use relaxed filtering heuristics. We do not use any multi-modal filtering but instead rely on the search engines to provide image-text correspondences. We remove text that is longer than 500 chars or formatted in JSON. We also remove images with an aspect ratio of more than 4 or covering less than 4096 pixels.
- We deduplicate all downloaded images using the Self-Supervised Descriptor for Image Copy Detection method (SSCD) (Pizzi et al., 2022). We keep the biggest image and collect all unique alt texts from all duplicates.
- We detect duplicates between the images and all evaluation datasets using the same SSCD method.

Our final dataset contains around 8.9M images and 11.8M alt texts, obtained from 125k queries, which is 71.2 images per query and 1.3 alt texts per image on average. We pay about $1,800 and $2,500 to download our Bing and Google subsets respectively. See Tab. 3 for an overview over the number of search queries, images and alt texts that make up our dataset and Appendix D for more details about the number and format of the requests issued to both search APIs.

## 4 EXPERIMENTAL SETUP

### 4.1 TRAINING

We train all models with the standard CLIP loss (Radford et al., 2021), a batch size of 2048, and random resized crop augmentation. Similar to Li et al. (2023b), we reduce the context size of the text encoder down from 77. We choose a length of 32 as a compromise to trade off speed and loss of data: On CC12M, 75% of captions have 32 tokens or less and are not affected. We train all models for 18 epochs using AdamW (Loshchilov & Hutter, 2019). Training on 9M images takes ∼30 hours on 8 RTX 2080 Ti GPUs with 11GB VRAM per GPU.

### 4.2 EVALUATED MODELS

On our LivingThings dataset, we train a ViT-B-32 model and randomly sample alt texts and knowledge graph labels as shown in Fig. 2. To compare to a similar-sized dataset, we also train models on CC12M. We download all available URLs, then detect and remove duplicates with the evaluation

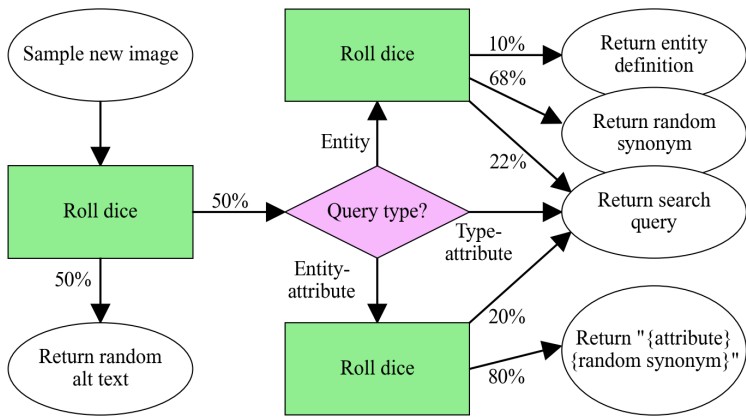

Figure 2: Flowchart for sampling text labels during training. 50% of the time we use an alt text, otherwise we sample a label from the knowledge graph.

datasets with the same procedure as detailed in Sec. 3.4. Then, we randomly split off a validation and test set with 20k images each and obtain 9.3M images. Additionally, we finetune the DataComp-1B model on our dataset to compare between pretraining and finetuning. Our training code is based on OpenCLIP (Ilharco et al., 2021).

In addition to the models trained on LivingThings and CC12M, we evaluate two other models with the same ViT-B-32 architecture: the original *OpenAI CLIP* (Radford et al., 2021), as well as a model pretrained on *DataComp-1B* (Gadre et al., 2023). To compare to models that are not only trained on more data but also have much more parameters, we evaluate a ViT-H-14 CLIP trained on *DFN-5B* (Fang et al., 2024a) and an EVA02-E-14+ model trained on *LAION-2B* (Fang et al., 2024b). We additionally compare with BioCLIP (Stevens et al., 2024), a ViT-B-16 model finetuned from OpenAI-CLIP on 10M biological image-text pairs. For more details on BioCLIP refer to Appendix F.

### 4.3 OBJECT CLASSIFICATION EVALUATION

To test the VLMs on object classification we use the same procedure as CLIP (Radford et al., 2021). Given an image $I$, class names $C_1, ..., C_N$, image encoder $f$ and text encoder $g$, we embed the image using the image encoder $\mathbf{v} = f(I)$. To acquire a text embedding for class $C_c$, the CLIP authors start by directly encoding the class names as $\mathbf{w}_c = g(C_c)$, e.g., "dog". Alternatively, they create several prompts $P$ using templates, e.g., "graffiti of a dog.", "a photo of the cool dog.", etc., then encode each prompt and compute the average embedding: $\mathbf{w}_c = \sum_{p \in P} g(p)/|P|$. They refer to this approach as using "context prompts". Finally, given the image and text embeddings, the prediction $p$ is the class which has the highest cosine similarity to the image. We evaluate all models on just encoding the class name, as well as on using the average embedding of the 80 context prompts that the CLIP author's used for ImageNet, and report the higher top-1 accuracy.

**Object datasets.** We evaluate on ImageNet (Deng et al., 2009), a popular image classification benchmark (Russakovsky et al., 2015). We use the ILSVRC2012 validation set, which contains 50,000 images from 1,000 classes. The classes include simple objects such as "broom", but also more fine-grained labels like 23 types of terrier dogs, e.g., "Staffordshire Bull Terrier". We split the classes into "living" (410 classes) and "other" (590 classes) using WordNet: Since ImageNet labels are built on WordNet nouns, we simply select all labels that are children of the node "Living Things" for the "living" set.

We also evaluate on two version of iNaturalist (Van Horn et al., 2018), a dataset for fine-grained species classification. The 2019 version contains 3,030 images in the validation set, each annotated with one of 1,010 Latin species. Similar to Parashar et al. (2023), we translate the species label to English via a combination of Wikidata SPARQL queries and manually looking up the species, and test models on both english and latin. The more challenging iNaturalist 2021 version contains 100k

images in the validation set and 10k different species. This version provides both Latin and English labels for the species.

We test on the CUB dataset, which contains 5,794 images in the original author's test set. Each image is annotated as one of 200 fine-grained species of birds, e.g., "grasshopper sparrow".

Finally, we evaluate on the RareSpecies dataset (Stevens et al., 2024), which requires models to not see the tested 400 species during training. We pretrain our model again and exclude all entities that appear in RareSpecies. We use various english and latin prompts for the name of the species and report the best result, for more details refer to Appendix G.

## 4.4 ATTRIBUTE CLASSIFICATION EVALUATION

The considered attribute benchmarks are framed as binary classification problems, as described by Bravo et al. (2023). A fixed set of attributes $A$ is annotated for each image as an attribute vector of length $A$ with value 0 if the attribute is not present, 1 if the attribute is present, and $-1$ for undecidable situations (e.g., when the attribute is "red-colored wing", but the photo only shows a bird's head). Models should assign high or low scores to positively or negatively labeled image-attribute pairs, respectively. Predictions on undecidable attributes do not influence the final score.

As metric, we use the established *mean Average Precision* (mAP) (Everingham et al., 2010); specifically, we use the implementation of Pedregosa et al. (2011). We compute AP per attribute and then average over all attributes. We only evaluate attribute classes which have at least one positive attribute, since otherwise AP cannot be computed for this attribute.

As in object classification, simply embedding only the attribute as text, e.g., "red", may not be the most efficient strategy. We consider using templates and synonyms to create a set of context prompts per attribute $P_a$. Then, we either create the average embedding as described in Sec. 4.3 or first evaluate the cosine similarities and report the prediction $p$ as the maximum similarity over all context prompts: $p = \max_{p_a \in P_a} \text{cossim}(f(I), g(p_a))$. In both attribute benchmarks, we evaluate all models on the "average embedding" and "maximum similarity" strategy and on a variety of context prompts depending on the dataset.

**Attribute datasets.** We evaluate on OVAD, which contains 80 object classes and 117 attribute classes. We consider the "oracle box" setting, where each input is one of 14,074 cropped bounding boxes annotated with one object class and an attribute vector of length 117. We group the dataset into animals, food, person, and other using the object label annotations. To create the prompt, we extend the prompts used by the original authors to include also class-specific prompts, instead of only class-agnostic prompts, see Appendix B.3 for details.

Additionally we evaluate attribute classification on CUB. We convert the given annotations into an image-attribute matrix with each attribute being annotated either negative, positive or unknown for each image, see Appendix A.1 for details. Then, we create prompts similar to OVAD, e.g., "bird has a striped pattern on its wing"., see Appendix A.4. This way, we match the setup of OVAD and can evaluate in the same way.

## 5 RESULTS

**Object classification.** Tab. 4 shows object classification results for various CLIP models. On iNaturalist 2021, we outperform models with 10-30 times more parameters and up to 100-500 times more pre-training images, despite never having seen any of the iNaturalist training images. On the highly contested ImageNet validation set, we match the performance of the original CLIP on our domain of animals and plants. Our model also performs well on CUB, distinguishing 200 bird species better than all other CLIP models of the same size. BioCLIP, trained specifically for organismal biology on a similar-sized dataset, is outperformed by our model on all benchmarks except the iNaturalist datasets that are included in BioCLIP's training set. This shows that our approach on training on a mix of entities, descriptions, attribute queries, and alt texts leads to a stronger and more general model, than aggregating biological datasets and training on a mix of entity names. In addition, the finetuning results show that our dataset can be used to improve existing CLIP models that were pre-trained on much larger datasets. To summarize, both our pretrained and finetuned models are able to do open-vocabulary recognition of fine-grained species on a variety of benchmarks.

Table 4: Object classification results. We mark the **best** and second best result. Results in *red* are not zero-shot, since the model has seen the training data.

| Dataset | Imgs (M) | Par. (M) | ImageNet 1k | Living | other | iNat. 2019 En | Lat | iNat. 2021 En | Lat | CUB | Rare species |
|---|---|---|---|---|---|---|---|---|---|---|---|
| # Classes → | | | 1,000 | 410 | 590 | 1,010 | 1,010 | 10k | 10k | 200 | 400 |
| LAION-2B EVA-E2-14+ | 2300.0 | **5045** | 82.0 | 85.2 | 80.9 | 24.5 | 12.4 | 22.3 | 8.0 | 84.9 | *49.2* |
| DFN-5B ViT-H-14 | 5000.0 | 986 | **83.4** | **85.4** | **83.2** | 28.3 | 31.4 | 25.1 | 23.9 | **88.1** | *52.9* |
| DataComp-1B | 1400.0 | 151 | 69.2 | 71.2 | 69.1 | 16.7 | 12.6 | 12.6 | 7.6 | 73.8 | *35.8* |
| OpenAI | 400.0 | 151 | 63.4 | 65.5 | 63.1 | 10.9 | 6.5 | 7.4 | 3.4 | 51.8 | *28.2* |
| CC12M | 9.3 | 151 | 32.2 | 30.0 | 35.4 | 2.1 | 0.6 | 0.9 | 0.1 | 9.7 | *9.2* |
| BioCLIP TreeOfLife-10M | 10.4 | 151 | 18.6 | 44.3 | 2.6 | *49.5* | *68.8* | *52.0* | *66.9* | 78.1 | 38.1 |
| Ours | 8.9 | 151 | 33.3 | 66.4 | 12.2 | 35.5 | 40.1 | 22.6 | 27.1 | 82.6 | **42.5** |
| Ours (finetuned) | 8.9 | 151 | 51.7 | 75.3 | 37.7 | **42.1** | **46.9** | 29.8 | 35.5 | 87.3 | *54.8* |

Table 5: Results for attribute classification on OVAD and CUB, mean Average Precision (mAP). We mark the **best** and second best result. Refer to Tab. 4 for dataset and model size.

| Dataset | OVAD animal | food | CUB All | Part Shape | Part Color | Part Pattern | Body Shape | Body Color | Body Size |
|---|---|---|---|---|---|---|---|---|---|
| Random Baseline | 28.4 | 31.4 | 11.4 | 17.6 | 9.6 | 19.8 | 7.14 | 11.9 | 20.0 |
| LAION-2B EVA-E2-14+ | **47.4** | 45.2 | **25.5** | 21.2 | 25.4 | **25.8** | 26.7 | 19.5 | **38.6** |
| DFN-5B ViT-H-14 | 47.1 | **48.1** | 24.9 | **21.7** | 24.5 | **25.8** | **28.0** | **20.0** | 38.4 |
| DataComp-1B | 45.1 | 45.1 | 25.4 | 20.4 | **25.6** | **25.8** | 22.8 | 18.3 | 37.6 |
| OpenAI | 46.0 | 43.7 | 23.5 | 20.1 | 23.3 | 23.6 | 26.8 | 17.9 | 35.1 |
| CC12M | 36.6 | 42.2 | 21.5 | 19.9 | 21.2 | 22.8 | 22.2 | 15.0 | 32.5 |
| BioCLIP TreeOfLife-10M | 33.0 | 36.5 | 17.3 | 20.8 | 16.2 | 21.8 | 20.3 | 10.6 | 23.9 |
| Ours | 40.9 | 41.5 | 24.1 | 21.3 | 23.8 | 24.2 | **28.0** | 18.5 | 35.0 |
| Ours (finetuned) | 40.2 | 41.5 | 24.2 | 21.4 | 23.9 | 24.2 | 26.3 | 19.2 | 36.2 |

**Attribute classification.** For this task, we show the results in Tab. 5. On CUB, we significantly improve over the same size CLIP model trained on CC12M and slightly outperform the original CLIP. Our model performs slightly below CLIP models that are trained on billions of images. We suspect that at this scale, models can transfer attribute knowledge between objects even better and therefore the general domain on which they are trained helps them improve over our animals and plants domain. For the categories "Part Shape" (e.g., "bird has a curved bill") and "Part Pattern" (e.g., "bird has a striped tail"), none of the models performs much better than the random baseline. This suggests that these tasks are extremely difficult to solve in a zero-shot manner. On other tasks, where either the attribute concerns the entire bird, or the question is about the color of a part, all models perform much better than the random baseline.

On OVAD, our model performs on par with or better than CC12M, but worse than the large CLIP models. Regarding the "food" category, our dataset covers plant species but not dishes, which make up 50% of the test set classes. Interestingly, the LAION model performs significantly better on the "animal" category than on the "food" category, while for the OpenAI model it is the other way round. We conduct a detailed error analysis, and find that many of the OVAD images have very low resolution, with not enough detail showing to decide many of the attributes. Our model performs significantly worse on these low-resolution images; see Appendix B.1 for details. We also find a significant number of wrong or doubtful ground-truth labels (e.g., a group of zebras labeled as "single", or a gray horse labeled as both "black" and "white" but not "gray"), which distort the evaluation results.

**Ablation studies.** In Tab. 6 we evaluate the mixture of alt text and knowledge graph labels we use during training. Notably, both training only on alt texts or only on knowledge graph labels consistently performs worse than our 50-50 mix. Removing all alt texts especially degrades the performance on the attribute benchmarks. This leads us to believe that much of the learned attribute knowledge comes from the alt texts. Next, we evaluate our choice of search queries and remove groups of queries to observe their contribution to the model quality. Both removing the WordNet queries and not using any of our LLM-generated attribute queries diminishes the model results.

Table 6: Ablation for our design choices. "Ours" refers to our default model with 50% alt texts and 50% labels obtained from the knowledge graphs, with a ViT-B-32 architecture trained on the full dataset. "No Attrs." means that we do not use any of the images found by using entity-attribute or type-attribute queries.

| Description | Imgs (M) | ImgNet Living | INat21 Eng. | INat21 Lat. | CUB Obj. | CUB Attr. | OVAD Animals | OVAD Food |
|---|---|---|---|---|---|---|---|---|
| Random Baseline | | 0.2 | 0.0 | 0.0 | 0.5 | 11.4 | 28.4 | 31.4 |
| Ours 0% alt texts | 8.9 | 62.6 | 21.4 | 25.6 | 81.0 | 20.1 | 38.5 | 37.5 |
| Ours 100% alt texts | 8.9 | 62.8 | 17.1 | 22.4 | 78.4 | 22.8 | 38.7 | 40.3 |
| Ours No Wordnet | 7.4 | 57.6 | 19.5 | 23.9 | 81.2 | 23.4 | 39.6 | 41.6 |
| Ours No Attrs. | 6.0 | 61.6 | 21.5 | 25.9 | 82.3 | 22.6 | 40.0 | 39.2 |
| Ours Google Only | 2.6 | 46.8 | 9.2 | 10.9 | 53.8 | 22.7 | 40.5 | **42.5** |
| Ours Bing Only | 6.9 | 64.1 | 20.8 | 24.6 | 81.3 | 23.8 | 38.5 | 42.0 |
| Ours | 8.9 | **66.4** | **22.6** | **27.1** | **82.6** | **24.1** | **40.9** | 41.5 |

Table 7: We compare various combinations of architecture and patch size.

| Description | Flops (G) | Par. (M) | ImgNet Living | INat21 Eng. | INat21 Lat. | CUB Obj. | CUB Attr. | OVAD Animals | OVAD Food |
|---|---|---|---|---|---|---|---|---|---|
| Random Baseline | | | 0.2 | 0.0 | 0.0 | 0.5 | 11.4 | 28.4 | 31.4 |
| Ours B-32 | 15 | 151 | 66.4 | 22.6 | 27.1 | 82.6 | **24.1** | 40.9 | 41.5 |
| Ours S-32 | 4 | 45 | 62.2 | 19.0 | 23.1 | 77.9 | 24.0 | 41.8 | 42.2 |
| Ours Ti-32 | 1 | 15 | 51.6 | 11.9 | 15.1 | 67.2 | 23.9 | 38.4 | 41.7 |
| Ours B-16 | 40 | 150 | **72.0** | **28.5** | **33.9** | **86.8** | 23.1 | 39.6 | 41.6 |
| Ours S-16 | 10 | 44 | 68.7 | 24.4 | 29.7 | 85.0 | 22.8 | **42.0** | 42.1 |
| Ours Ti-16 | 3 | 14 | 58.4 | 16.3 | 20.5 | 76.2 | 23.9 | 40.7 | **42.8** |

Regarding the search engines, upon manual inspection we find that the search results of the Google API are significantly worse than the results of the Bing API. This is consistent with the performance differences between training only on Bing results and only on Google results.

In Tab. 7 we compare different architecture choices with the three different sizes base, small and tiny, as well as two different patch sizes 32x32px and 16x16px. Note that smaller patch sizes are significantly more expensive to traindue to a quadratically higher token count. For object classification, the biggest model with the smallest patch size is best. For attribute classification, the picture is again less clear with the best performing model changing between benchmarks. Nonetheless, we find that it might be worth trading off a smaller model against a smaller patch size.

## 6 CONCLUSIONS

We have developed a compact high-quality dataset LivingThings consisting of 8.9M images from the domain of animals and plants, paired with 12.2M texts. Our method is generic, leveraging knowledge graphs (WordNet and Wikidata) and image search engines, and can be applied to any domain covered by the given knowledge graphs. We have demonstrated that we can train a CLIP model on our dataset with little compute, yet with a performance that is comparable to or even better than that of much more expensive-to-train models. By adapting the CUB benchmark to VLM-style evaluation, we also enable more comprehensive assessment in the field of open-vocabulary attribute understanding.

While previous studies have shown that alt texts offer better supervision than the search queries used to find images, in our work, we demonstrate that combining alt texts with search queries can indeed improve performance compared to using alt texts alone. This approach opens up new possibilities for supervision in VLM training. Ultimately, we hope to provide researchers with a useful tool for building custom datasets and facilitating affordable VLM pretraining experiments.

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

# A  DETAILS AND ADDITIONAL RESULTS FOR CUB

## A.1  CONVERTING WORKER ANNOTATIONS TO ATTRIBUTE LABELS

Wah et al. (2011) design a vocabulary of 28 attribute groupings and 312 binary attributes, e.g., "wing color" with 15 color choices. They create the attribute labels via crowd sourced workers such that for each image-attribute combination, there is exactly one worker annotation with a binary label whether the attribute is present, and a certainty score with values (1, not visible), (2, guessing), (3, probably), (4, definitely). We now convert these ratings into image-attribute labels with values 0, 1 and -1 meaning attribute is not present, attribute is present, and attribute is unknown, respectively, same as in the OVAD dataset. The average certainty is 3.3. We consider all ratings with certainty (1, not visible) and (2, guessing) as unknown and otherwise use the annotated label.

## A.2  RESULTS

In Supp. Tab. 8 we provide additional results for CUB attribute classification. Specifically, we show the model and ablation results for various attribute groups.

Table 8: CUB attribute classification, mean Average Precision (mAP). We select the best prompt and evaluation setup per model. "Ours" refers to our default model with 50% alt texts and 50% labels obtained from the knowledge graphs, with a ViT-B-32 architecture trained on the full dataset. We mark the **best** and second best result. Refer to Supp. Tab. 9 for model GFlops.

| Dataset | Imgs (K) | Par. (M) | All | Part Shape | Part Color | Part Pattern | Body Shape | Body Color | Body Size |
|---|---|---|---|---|---|---|---|---|---|
| Num. attributes → | | | 312 | 23 | 224 | 31 | 14 | 15 | 5 |
| Num. pos. labels (K) → | | | 169K | 17K | 104K | 29K | 5K | 10K | 5K |
| Num. neg. labels (K) → | | | 1,336K | 83K | 975K | 119K | 67K | 72K | 20K |
| Random Baseline | | | 11.4 | 17.6 | 9.6 | 19.8 | 7.14 | 11.9 | 20.0 |
| Ours 0% alt texts | 8.9 | 151 | 20.1 | 21.9 | 19.0 | 22.5 | 23.2 | 16.4 | 30.6 |
| Ours 100% alt texts | 8.9 | 151 | 22.8 | 21.6 | 22.2 | 23.1 | 28.1 | 18.3 | 35.2 |
| Ours No Wordnet | 7.4 | 151 | 23.4 | 21.6 | 22.8 | 23.6 | 28.5 | **20.0** | 35.8 |
| Ours No Attrs. | 6.0 | 151 | 22.6 | 20.8 | 22.0 | 24.6 | 28.6 | 17.8 | 32.1 |
| Ours No Attr.-Noun | 6.8 | 151 | 23.4 | 21.3 | 23.1 | 24.0 | 26.6 | 17.8 | 34.1 |
| Ours Google Only | 2.6 | 151 | 22.7 | **22.0** | 22.2 | 23.1 | 25.6 | 18.5 | 34.4 |
| Ours Bing Only | 6.9 | 151 | 23.8 | 21.6 | 23.5 | 23.4 | **28.8** | 18.6 | 35.2 |
| Ours S-32 | 8.9 | 45 | 24.0 | 21.9 | 23.8 | 23.9 | 25.7 | 18.7 | 36.1 |
| Ours Ti-32 | 8.9 | 15 | 23.9 | 21.9 | 23.7 | 24.1 | 27.5 | 18.4 | 34.4 |
| Ours B-16 | 8.9 | 150 | 23.1 | 21.3 | 22.5 | 24.4 | 26.6 | 18.1 | 35.6 |
| Ours S-16 | 8.9 | 44 | 22.8 | 21.3 | 22.0 | 24.5 | 28.2 | 18.6 | 35.2 |
| Ours Ti-16 | 8.9 | 14 | 23.9 | 21.1 | 23.7 | 24.5 | 27.0 | 18.2 | 35.5 |
| LAION-2B EVA-E2-14+ | 2300.0 | 5045 | **25.5** | 21.2 | 25.4 | **25.8** | 26.7 | 19.5 | **38.6** |
| DFN-5B ViT-H-14 | 5000.0 | 986 | 24.9 | 21.7 | 24.5 | **25.8** | 28.0 | **20.0** | 38.4 |
| DataComp-1B | 1400.0 | 151 | 25.4 | 20.4 | **25.6** | **25.8** | 22.8 | 18.3 | 37.6 |
| OpenAI | 400.0 | 151 | 23.5 | 20.1 | 23.3 | 23.6 | 26.8 | 17.9 | 35.1 |
| CC12M B-32 | 9.3 | 151 | 21.5 | 19.9 | 21.2 | 22.8 | 22.2 | 15.0 | 32.5 |
| BioCLIP TreeOfLife-10M | 10.4 | 150 | 17.3 | 20.8 | 16.2 | 21.8 | 20.3 | 10.6 | 23.9 |
| Ours | 8.9 | 151 | 24.1 | 21.3 | 23.8 | 24.2 | 28.0 | 18.5 | 35.0 |
| Ours (finetuned) | 8.9 | 151 | 24.2 | 21.4 | 23.9 | 24.2 | 26.3 | 19.2 | 36.2 |

## A.3  ATTRIBUTE GROUPS

We sort the CUB attribute groups into categories. Statistics about these groups can be found in the header of Supp. Tab. 8. **part shape**: bill shape, tail shape, bill length, wing shape, **part color**: wing color, upperparts color, underparts color, back color, upper tail color, breast color, throat color, eye color, forehead color, under tail color, nape color, belly color, leg color, bill color, crown color, **part**

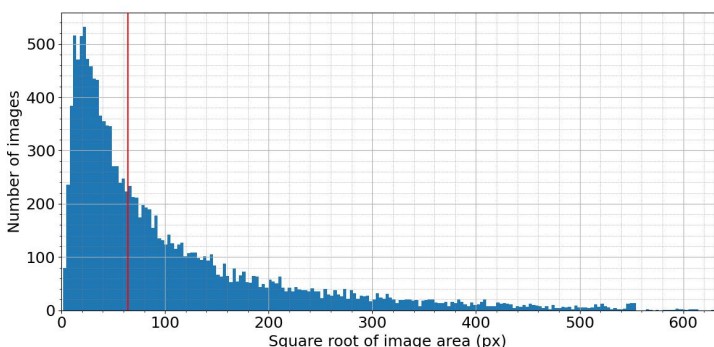

Figure 3: OVAD image size distribution. The red line indicates an image area of $64^2$px.

**pattern**: breast pattern, head pattern, back pattern, tail pattern, belly pattern, wing pattern, **primary**: size, shape, primary color

### A.4    PROMPTS

In order to evaluate the CUB attribute labels, we need reliable prompts for each attribute. We create separate templates for each of the categories defined in Appendix A.3. Then, to cover a large area of possible prompts, we consider the original prompting strategies by Bravo et al. (2023) to vary the articles, prepositions and nouns. For nouns, we choose "bird", "animal" and the empty string. We end up with a diverse set of 37 prompt settings, each with a median of 24 prompts per attribute. Example prompts: "bird has a purple-colored nape", "the animal with its wing in a long shape", or "red-colored under tail". To create a single model prediction from several prompts prompts, we evaluate both the "average embedding" (Sec. 4.3) and "maximum similarity" (Sec. 4.4) strategies to create model the prediction.

## B    DETAILS AND ADDITIONAL RESULTS FOR OVAD

### B.1    INPUT RESOLUTION ANALYSIS

We find that the cropped boxes in the OVAD oracle box task are quite small and show the image size distribution in Supp. Fig. 3. To analyze if this influences model behaviour, we then filter OVAD and only keep boxes with an area of at least 64x64 pixels. This discards roughly half the boxes. We compare behaviour of models on the original boxes in Supp. Tab. 9 and the filtered boxes in Supp. Tab. 10. Most models slightly increase their performance, though, the biggest models increase less than our model. They may have seen more blurry or low-resolution images and can understand them better.

### B.2    DATASET GROUPING

We sort the OVAD boxes into the following groups:

**food**: banana, apple, sandwich, orange, broccoli, carrot, hot dog, pizza, donut, cake. **animal**: bird, cat, dog, horse, sheep, cow, elephant, bear, zebra, giraffe. **person**: person. **other**: bicycle, car, motorcycle, airplane, bus, train, truck, boat, traffic light, fire hydrant, stop sign, parking meter, bench, backpack, umbrella, handbag, tie, suitcase, frisbee, skis, snowboard, sports ball, kite, baseball bat, baseball glove, skateboard, surfboard, tennis racket, bottle, wine glass, cup, fork, knife, spoon, bowl, chair, couch, potted plant, bed, dining table, toilet, tv, laptop, mouse, remote, keyboard, cell phone, microwave, oven, toaster, sink, refrigerator, book, clock, vase, scissors, teddy bear, hair drier, toothbrush

Table 9: Results for OVAD attribute classification, mean Average Precision (mAP). For each model and group, we select the best prompt and embedding strategy. Gray: Our model is out of distribution. We mark the **best** and second best result.

| Dataset | Imgs (M) | Flops (G) | Par. (M) | OVAD | | | | |
| | | | | all | animal | food | person | other |
| --- | --- | --- | --- | --- | --- | --- | --- | --- |
| Num. instances → | | | | 14,074 | 1,239 | 1,349 | 4,116 | 7,370 |
| Num. attributes → | | | | 116 | 44 | 30 | 53 | 65 |
| Num. pos. labels (K) → | | | | 122 | 11 | 10 | 32 | 69 |
| Num. neg. labels (K) → | | | | 1,248 | 26 | 23 | 107 | 255 |
| Random Baseline | | | | 8.7 | 28.4 | 31.4 | 21.9 | 22.3 |
| Ours 0% alt texts | 8.9 | 15 | 151 | 12.7 | 38.5 | 37.5 | 23.7 | 25.2 |
| Ours 100% alt texts | 8.9 | 15 | 151 | 13.1 | 38.7 | 40.3 | 26.5 | 26.8 |
| Ours No Wordnet | 7.4 | 15 | 151 | 12.8 | 39.6 | 41.6 | 25.9 | 26.6 |
| Ours No Attrs. | 6.0 | 15 | 151 | 12.7 | 40.0 | 39.2 | 25.5 | 26.2 |
| Ours No Attr.-Noun | 6.8 | 15 | 151 | 12.8 | 40.3 | 41.6 | 25.5 | 26.1 |
| Ours Google Only | 2.6 | 15 | 151 | 12.9 | 40.5 | 42.5 | 25.9 | 26.3 |
| Ours Bing Only | 6.9 | 15 | 151 | 12.8 | 38.5 | 42.0 | 25.5 | 26.2 |
| LAION-2B EVA-E2-14+ | 2300.0 | 2331 | 5045 | 18.1 | **47.4** | 45.2 | 34.3 | 30.7 |
| DFN-5B ViT-H-14 | 5000.0 | 370 | 986 | **22.3** | 47.1 | **48.1** | **36.6** | **34.4** |
| DataComp-1B | 1400.0 | 15 | 151 | 17.7 | 45.1 | 45.1 | 32.6 | 31.0 |
| OpenAI | 400.0 | 15 | 151 | 16.4 | 46.0 | 43.7 | 32.0 | 30.5 |
| CC12M | 9.3 | 15 | 151 | 13.3 | 36.6 | 42.2 | 28.0 | 27.0 |
| BioCLIP TreeOfLife-10M | 10.4 | 40 | 150 | 10.3 | 33.0 | 36.5 | 23.0 | 23.5 |
| Ours | 8.9 | 15 | 151 | 13.1 | 40.9 | 41.5 | 25.6 | 27.3 |
| Ours (finetuned) | 8.9 | 15 | 151 | 15.0 | 40.2 | 41.5 | 27.8 | 27.8 |

## B.3 OVAD PROMPTS

When evaluating different prompts for attribute classification, we start with the original prompting strategies by Bravo et al. (2023). We create all permutations of the following choices. To create a single model prediction from several prompts, we evaluate both the "average embedding" (Sec. 4.3) and "maximum similarity" (Sec. 4.4) strategies to create model the prediction. To write the prompt, we use either "a", "the", or no article in the prompt and the word "object" or an empty string as the noun. Additionally, we use the entity name directly ("a small sheep") or the natural type ("a small animal"). In total, we evaluate 48 prompt settings per model and report the best result. Models are quite sensitive to the exact prompt setting. We show this in Supp. Fig. 4 and Supp. Fig. 5 by plotting the distribution of the 48 different results for our model as well as two models trained on DataComp-1B and CC12M, respectively.

## C QUALITATIVE EXAMPLES OF OUR DATASET

We show randomly sampled images and corresponding textual information of our dataset in Supp. Fig. 6 and Supp. Fig. 7

## D IMAGE SEARCH APIs

**Google** The Google Image Search API is available via the Google Cloud Platform, and requires an existing programmable search engine to function. It returns up to 10 images per request and page (e.g. page 1 corresponds to images 1 to 10, page 2 to images 11 to 20, and so on). However, one can only get results for the first 10 pages, or the top 100 images, and only issue 10,000 API calls per day. It costs 5$ per 1,000 API calls, resulting in costs of about 500$ to download 1M images. We found the search results from the Google API to be quite different, and arguably worse, from the ones returned when using the regular Google image search. For all our API requests we set the parameter *imgColorType* to *color*, *imgType* to *photo*, *lr* to *lang_en*, and *excludeTerms* to *drawing clipart illustration cartoon vector painting*. This way we get mostly real-world images in our search

Table 10: Results for OVAD attribute classification, mean Average Precision (mAP). *In this table, we remove all box instances that have an area smaller than $64^2 px$.* For each model and group, we select the best prompt and embedding strategy. Gray: Our model is out of distribution. We mark the **best** and second best result.

| Dataset | Imgs (M) | Flops (G) | Par. (M) | OVAD all | animal | food | person | other |
|---|---|---|---|---|---|---|---|---|
| Num. instances → | | | | 6,933 | 615 | 680 | 2,011 | 3,627 |
| Num. attribute classes → | | | | 115 | 41 | 30 | 52 | 65 |
| Num. pos. labels → | | | | 60K | 6K | 5K | 15K | 34K |
| Num. neg. labels → | | | | 609K | 12K | 12K | 51K | 126K |
| Random Baseline | | | | 8.8 | 30.5 | 31.4 | 22.3 | 22.3 |
| Ours 0% alt texts | 8.9 | 15 | 151 | 13.1 | 41.3 | 38.9 | 24.6 | 25.4 |
| Ours 100% alt texts | 8.9 | 15 | 151 | 13.4 | 40.8 | 41.2 | 28.2 | 27.1 |
| Ours No Wordnet | 7.4 | 15 | 151 | 13.0 | 43.1 | 41.8 | 27.1 | 26.7 |
| Ours No Attrs. | 6.0 | 15 | 151 | 13.1 | 40.2 | 40.2 | 26.6 | 26.7 |
| Ours No Attr.-Noun | 6.8 | 15 | 151 | 13.3 | 45.6 | 42.1 | 26.5 | 26.4 |
| Ours Google Only | 2.6 | 15 | 151 | 13.2 | 38.4 | 43.4 | 26.8 | 26.7 |
| Ours Bing Only | 6.9 | 15 | 151 | 13.0 | 42.4 | 42.3 | 26.6 | 26.6 |
| LAION-2B EVA-E2-14+ | 2300.0 | 2331 | 5045 | 18.5 | 47.0 | 46.7 | 35.1 | 31.3 |
| DFN-5B ViT-H-14 | 5000.0 | 370 | 986 | **23.0** | **47.9** | **48.8** | **38.1** | **35.1** |
| DataComp-1B | 1400.0 | 15 | 151 | 18.2 | 44.7 | 46.9 | 33.4 | 31.6 |
| OpenAI | 400.0 | 15 | 151 | 16.9 | 45.1 | 42.9 | 33.5 | 31.0 |
| CC12M | 9.3 | 15 | 151 | 13.9 | 37.4 | 42.9 | 29.1 | 27.4 |
| BioCLIP TreeOfLife-10M | 10.4 | 40 | 150 | 10.5 | 38.1 | 36.8 | 23.7 | 23.7 |
| Ours | 8.9 | 15 | 151 | 13.5 | 43.7 | 43.2 | 26.2 | 27.7 |
| Ours (finetuned) | 8.9 | 15 | 151 | 15.5 | 44.3 | 41.9 | 28.7 | 28.3 |

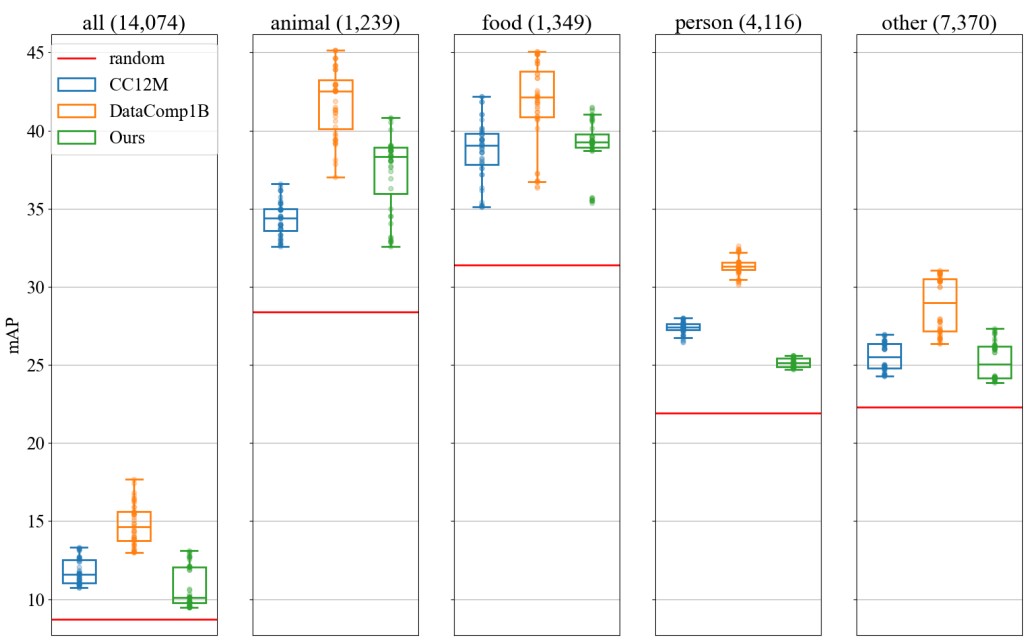

Figure 4: Model performance on OVAD when using different prompts.

results. We additionally add all aliases, taxon common names, taxon names, and the natural type of the sought entity to the *orTerms* parameter for entity and entity-attribute queries. Because the Google API returns only up to 10 images per request and page, we search for the following number of pages

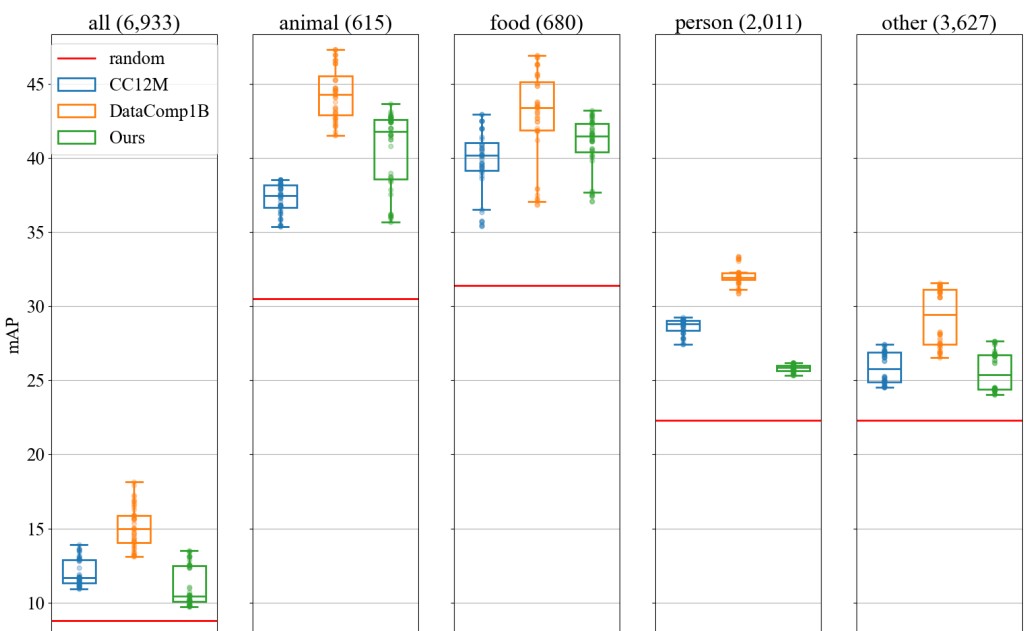

Figure 5: Model performance on OVAD when using different prompts. Here, we remove all box instances that have an area smaller than $64^2$px.

for each query in the respective query sets: 10 pages each for all 17k WordNet entity queries, 2 pages each for the top 28k Wikidata entity queries for both animals and plants, 4 pages each for all Wikidata entity-attribute queries, and 10 pages each for all Wikidata type-attribute queries. In total, this amounts about 500k API requests, costing us around 2,500\$.

**Bing** The Bing Image Search API is available via Microsoft Azure. It returns up to 150 images per request and has no restrictions on the number of accessible pages. It limits the number of requests to 100 per second and costs 18\$ per 1,000 API calls, resulting in costs of about 120\$ to download 1M images. In our experience, the returned images closely match the ones from the regular Bing image search. For all our API requests we set the parameter *imageType* to *Photo* and *color* to *ColorOnly*. Unlike the Google API, Bing does not have a way to specify *orTerms* via a separate request parameter, so we just add the natural type of the sought entity to the search query directly, in the case of entity queries ("tomato plant") and entity-attribute queries ("sitting orangutan animal"). Because the Bing API returns 150 images per request, we make only one request for each of the following queries: all WordNet entity queries, the top 20k Wikidata entity queries for both animals and plants, all Wikidata entity-attribute queries, and all Wikidata type-attribute queries. In total, this amounts to about 100k API requests, costing us around 1,800\$. Considering that we end up with 6.9M images from Bing and 2.3M images from Google, we find the Bing Search API to have a much better value for money ratio.

## E  WIKIDATA SPARQL QUERIES USED TO BUILD THE DATASET

To query the Wikidata knowledge graph, we use the QLever SPARQL engine (Bast & Buchhold, 2017). We get our list of animals with the SPARQL query shown in Supp. Fig. 8. The important part is the *UNION* expression at the beginning of the query, defining an animal to be either a subclass of the animal entity or a child taxon of it. In SPARQL this can be expressed via

```
?ent (wdt:P31/wdt:P279*)wdt:P279+ wd:Q729
```

for the subclass relation, and via

```
?ent wdt:P171+ wd:Q729
```

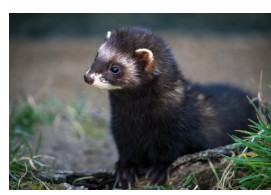
Wikidata entity + attribute
running polecat
Bing

A wonderful polecat in its woodland surroundings / Polecats Unveiled: Sleek Predators in the Countryside (Mustela Putorius) - Glenlivet Wildlife / Polecats Unveiled: Sleek Predators in the Countryside (Mustela Putorius) / Black Polecat Photos and Premium High Res Pictures - Getty Images / Do Cats Eat Ferrets – What You Should Know! – FAQcats.com / Do Cats Eat Ferrets – What You Should Know!

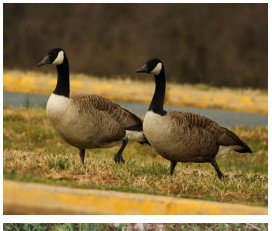
Wikidata entity + attribute
summer Canada goose
Bing

Canada Geese Goose Branta - Free photo on Pixabay - Pixabay / Canada Geese Goose Branta · Free photo on Pixabay / Facts about geese / Canada Geese Goose Branta Free Photo On Pixabay Pixabay, 45% OFF

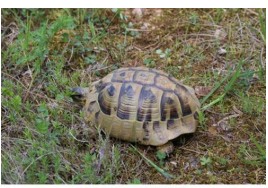
Wikidata entity + attribute
wild tortoise
Google

Greek Tortoise Testudo Graeca Hiding Shell Stock Photo 1425661328 — Shutterstock / Elongated Tortoise Indotestudo Elongata Yellow Tortoise Stock Photo 1463951543 — Shutterstock

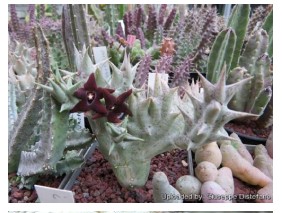
Wikidata entity
Orbea decaisneana
Google

Orbea decaisneana subs. hesperidum f. cristata

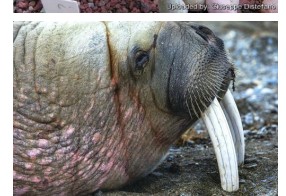
Wikidata entity + attribute
old walrus
Bing

What Is A Walrus? / What is a Walrus - Walrus Habitat and Behavior - Wild Focus Expeditions / Portrait of an old bull walrus resting on his teeth, tooth walker

Figure 6: Randomly sampled images from our dataset together with the corresponding query set, search query, search API, and alt texts (separated by /).

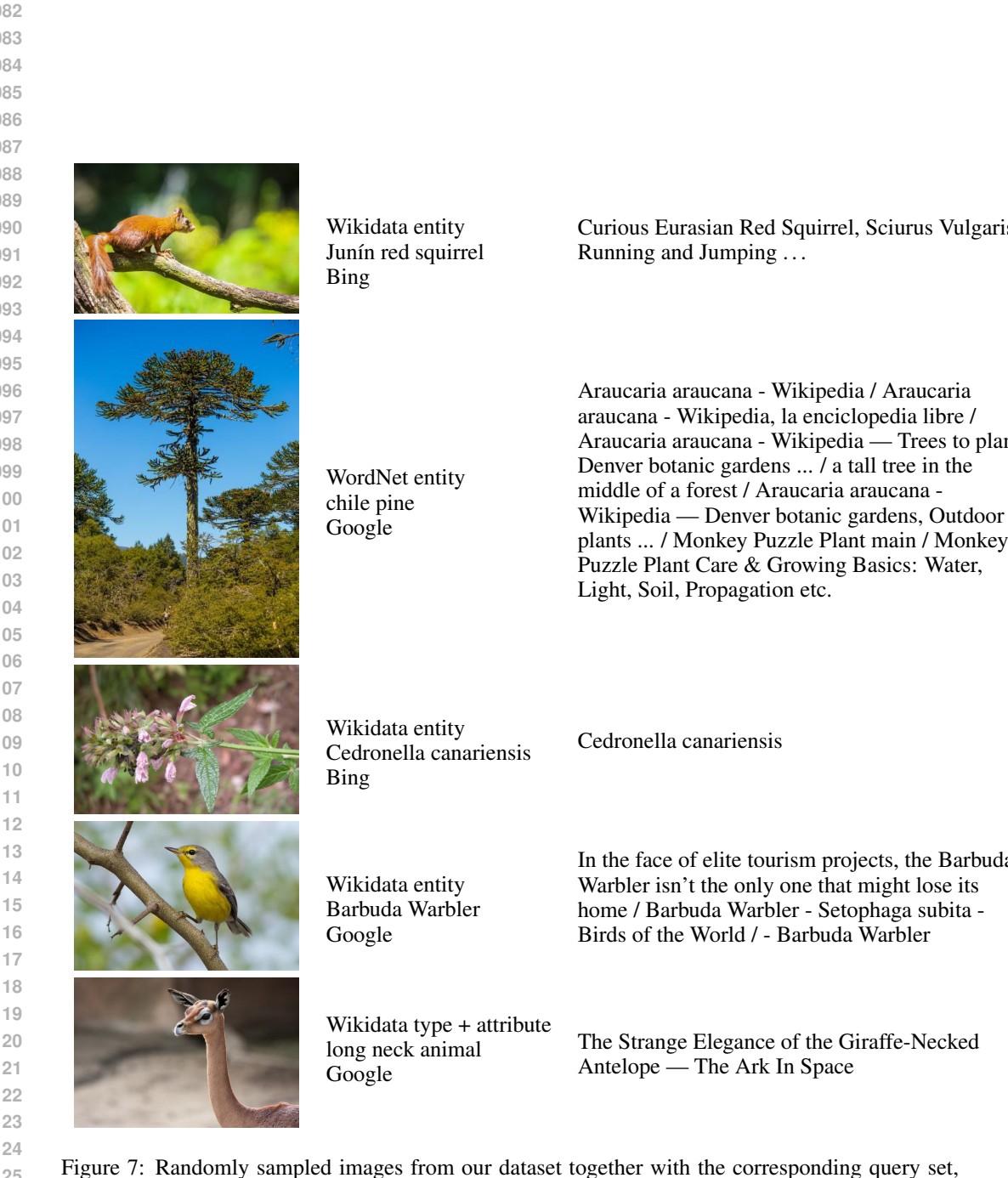

| | Wikidata entity
Junín red squirrel
Bing | Curious Eurasian Red Squirrel, Sciurus Vulgaris, Running and Jumping . . . |
| | WordNet entity
chile pine
Google | Araucaria araucana - Wikipedia / Araucaria araucana - Wikipedia, la enciclopedia libre / Araucaria araucana - Wikipedia — Trees to plant, Denver botanic gardens ... / a tall tree in the middle of a forest / Araucaria araucana - Wikipedia — Denver botanic gardens, Outdoor plants ... / Monkey Puzzle Plant main / Monkey Puzzle Plant Care & Growing Basics: Water, Light, Soil, Propagation etc. |
| | Wikidata entity
Cedronella canariensis
Bing | Cedronella canariensis |
| | Wikidata entity
Barbuda Warbler
Google | In the face of elite tourism projects, the Barbuda Warbler isn't the only one that might lose its home / Barbuda Warbler - Setophaga subita - Birds of the World / - Barbuda Warbler |
| | Wikidata type + attribute
long neck animal
Google | The Strange Elegance of the Giraffe-Necked Antelope — The Ark In Space |

Figure 7: Randomly sampled images from our dataset together with the corresponding query set, search query, search API, and alt texts (separated by /).

```
1134   PREFIX schema: <http://schema.org/>
1135   PREFIX wikibase: <http://wikiba.se/ontology#>
1136   PREFIX wdt: <http://www.wikidata.org/prop/direct/>
1137   PREFIX wd: <http://www.wikidata.org/entity/>
1138   PREFIX rdfs: <http://www.w3.org/2000/01/rdf-schema#>
1139   PREFIX skos: <http://www.w3.org/2004/02/skos/core#>
1140   SELECT
1141   ?ent
1142   ?label
1143   ?desc
1144   ?links
1145   (GROUP_CONCAT(DISTINCT ?alias; SEPARATOR=";") AS ?aliases)
1146   (GROUP_CONCAT(DISTINCT ?common_name; SEPARATOR=";") AS ?common_names)
1147   (GROUP_CONCAT(DISTINCT ?taxon_name; SEPARATOR=";") AS ?taxon_names)
1148   (GROUP_CONCAT(DISTINCT ?image; SEPARATOR=";") AS ?images)
1149   WHERE {
1150       # subclass of animal
1151       { ?ent (wdt:P31/wdt:P279*)|wdt:P279+ wd:Q729 . }
1152       UNION
1153       # child taxon of animal
1154       { ?ent wdt:P171+ wd:Q729 . }
1155       # filter out humans
1156       MINUS { ?ent (wdt:P31/wdt:P279*)|wdt:P279+ wd:Q5 . }
1157       # filter out mythical creatures
1158       MINUS { ?ent (wdt:P31/wdt:P279*)|wdt:P279+ wd:Q24334299 }
1159       # filter out individuals, e.g. named animals like Krake Paul
1160       MINUS { ?ent (wdt:P31/wdt:P279*)|wdt:P279+ wd:Q795052 }
1161       ?ent @en@rdfs:label ?label .
1162       OPTIONAL { ?ent ^schema:about/wikibase:sitelinks ?links }
1163       OPTIONAL { ?ent @en@schema:description ?desc }
1164       OPTIONAL { ?ent @en@skos:altLabel ?alias }
1165       OPTIONAL { ?ent @en@wdt:P1843 ?common_name }
1166       OPTIONAL { ?ent wdt:P225 ?taxon_name }
1167       ?ent wdt:P18 ?image
1168   }
1169   GROUP BY ?ent ?label ?desc ?links
1170   ORDER BY DESC(?links)
```

Figure 8: Our SPARQL query for extracting animals from Wikidata

for the child taxon relation. Here, `?ent` is the entity we want to retrieve, `wdt:P31` means *instance of*, `wdt:P279` means *subclass of*, `wdt:P171` means *parent taxon*, and `wd:Q729` is the animal entity. We use the modifiers * and + to also allow relational paths of length *zero or more* or *one or more* from an entity to the animal entity.

Similarly, Supp. Fig. 9 shows the query we use to extract plants and fruits. We include a second *UNION* expression to explicitly take fruits into account. Other than that, this query follows the same structure as the animal query.

```
PREFIX schema: <http://schema.org/>
PREFIX wikibase: <http://wikiba.se/ontology#>
PREFIX wdt: <http://www.wikidata.org/prop/direct/>
PREFIX wd: <http://www.wikidata.org/entity/>
PREFIX rdfs: <http://www.w3.org/2000/01/rdf-schema#>
PREFIX skos: <http://www.w3.org/2004/02/skos/core#>
SELECT
?ent
?label
?desc
?links
(GROUP_CONCAT(DISTINCT ?alias; SEPARATOR=";") AS ?aliases)
(GROUP_CONCAT(DISTINCT ?common_name; SEPARATOR=";") AS ?common_names)
(GROUP_CONCAT(DISTINCT ?taxon_name; SEPARATOR=";") AS ?taxon_names)
(GROUP_CONCAT(DISTINCT ?image; SEPARATOR=";") AS ?images)
WHERE {
    # subclass of plant
    { ?ent (wdt:P31/wdt:P279*)|wdt:P279+ wd:Q756 . }
    UNION
    # child taxon of plant
    { ?ent wdt:P171+ wd:Q756 . }
    # fruits
    UNION
    {
        { ?taxon (wdt:P31/wdt:P279*)|wdt:P279+ wd:Q756 . }
        UNION
        { ?taxon wdt:P171+ wd:Q756 . }
        { ?ent wdt:P1582 ?taxon . }
        UNION
        { ?taxon wdt:P1672 ?ent . }
        ?ent wdt:P31|wdt:P279 ?fruit .
        VALUES ?fruit { wd:Q3314483 wd:Q1364 }
    }
    # filter out cultivars
    MINUS { ?ent (wdt:P31/wdt:P279*)|wdt:P279+ wd:Q4886 }
    # filter out individuals, e.g. memorable trees
    MINUS { ?ent (wdt:P31/wdt:P279*)|wdt:P279+ wd:Q795052 }
    # filter out all plants that have a coordinate location
    MINUS { ?ent wdt:P625 ?coord }
    ?ent @en@rdfs:label ?label .
    OPTIONAL { ?ent ^schema:about/wikibase:sitelinks ?links }
    OPTIONAL { ?ent @en@schema:description ?desc }
    OPTIONAL { ?ent @en@skos:altLabel ?alias }
    OPTIONAL { ?ent @en@wdt:P1843 ?common_name }
    OPTIONAL { ?ent wdt:P225 ?taxon_name }
    ?ent wdt:P18 ?image
}
GROUP BY ?ent ?label ?desc ?links
ORDER BY DESC(?links)
```

Figure 9: Our SPARQL query for extracting plants and fruits from Wikidata

Table 11: Evaluation on the Rare Species dataset. Models in *red* have potentially seen the rare species during training. We evaluate different types of class name texts.

| Dataset | Imgs (M) | Par. (M) | Common | Scientific | Scientific + common | Taxonomic | Taxonomic + common |
|---|---|---|---|---|---|---|---|
| LAION-2B EVA-E2-14+ | 2300.0 | 5045 | *49.2* | *14.4* | *45.8* | *18.3* | *48.3* |
| DFN-5B ViT-H-14 | 5000.0 | 986 | *51.6* | *32.4* | *52.9* | *33.8* | *51.4* |
| DataComp-1B | 1400.0 | 151 | *35.8* | *13.3* | *35.1* | *13.3* | *35.0* |
| OpenAI | 400.0 | 151 | *28.2* | *9.3* | *27.6* | *9.9* | *26.7* |
| CC12M | 9.3 | 151 | *9.2* | *0.9* | *6.6* | *1.4* | *3.7* |
| Ours | 8.9 | 151 | *46.0* | *46.0* | *48.3* | *37.3* | *41.5* |
| BioCLIP TreeOfLife-10M | 10.4 | 150 | 31.8 | 30.2 | 37.1 | 34.1 | 38.1 |
| Ours (no rare species seen) | 8.9 | 151 | 38.8 | 36.1 | **42.5** | 32.3 | 36.7 |

# F BioCLIP DETAILS

Stevens et al. (2024) aim to create a general vision model for organismal biology and curate the TreeOfLife-10M dataset, which is based on the sources outlined below. Encyclopedia of Life (EOL, 2018) is a project aimed to aggregate biological knowledge in an online encyclopedia. Stevens et al. (2024) download 6.6M images of 440K species from EOL. They also incorporate iNaturalist 2021 (Van Horn et al., 2018) described in Sec. 4.3 and add the 2.7M images of 10K species from the training set. Finally, they add BIOSCAN-1M (Gharaee et al., 2023) which contains 1M lab images from 494 different insect families. Their model BioCLIP is trained on a mix of common names (e.g., "Rufous-crowned Sparrow"), scientific names (e.g., "Aimophila ruficeps"), as well as taxonomic names (the full taxonomy from kingdom to species, e.g., "Animalia Chordata Aves Passeriformes Passerellidae Aimophila ruficeps"). They finetune the OpenAI ViT-B-16 model for 100 epochs on 16 A100 80GB GPUs.

# G DETAILED RESULTS FOR RARE SPECIES

The "Rare Species" benchmark proposed by Stevens et al. (2024) consists of 400 species with 30 images each and is used to test generalization to unseen taxa. The species are deliberately removed from TreeOfLife-10M, such that BioCLIP has not seen them during training. We replicate this process and find 290 of the 400 species as entities in our dataset using substring matching. We then remove these 290 entities from our LivingThings dataset and train our baseline model again.

As class names we evaluate all text types proposed by Stevens et al. (2024), i.e, various combinations of the latin taxonomy and the english common name. Same as in Sec. 4.3 we evaluate on both the CLIP ImageNet prompt and no prompt, and report the better of both accuracies.

Our model significantly outperforms BioCLIP in classifying unseen rare species as shown in Supp. Table 11. It has the highest accuracy when given mixed scientific and common class names, which is to be expected, since it has seen those types of text during training, but has not seen full taxonomic class names. BioCLIP performs best on a mix of taxonomic and common names, however it does not match the performance of our model.

# H SCALING STUDY

In Supp. Fig. 10 and Supp. Tab. 12 we train a separate model for various scales of our dataset. Each time the dataset size is doubled, we get a significantly better model, and there is no saturation at the biggest scale.

We also observe unexpected performance changes in both OVAD categories. Upon inspection we find that these are artifacts coming from the small size of the food and animal category with around 1k datapoints each. E.g., the food category has only 15 positive attributes for the attribute "gray". The 1.6% and 6.2% size models learn quickly to predict these simple attributes well, which gives them a huge boost in mean Average Precision. But, there are actually not enough labels per attribute to draw meaningful conclusions. In the bigger CUB attribute benchmark, these effects vanish.

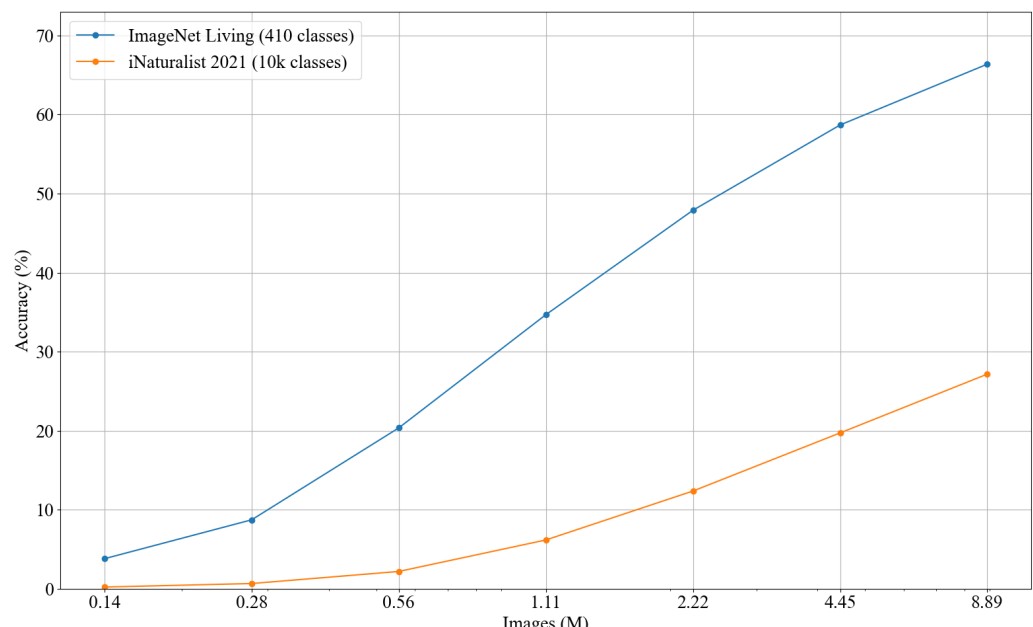

Figure 10: Visualization of model performance when scaling the dataset size.

Table 12: Model performance when scaling the dataset size.

| Description | Imgs (M) | ImgNet Living | INat21 Eng. | INat21 Lat. | CUB Obj. | CUB Attr. | OVAD Animals | OVAD Food |
|---|---|---|---|---|---|---|---|---|
| Random Baseline | | 0.2 | 0.0 | 0.0 | 0.5 | 11.4 | 28.4 | 31.4 |
| Ours (1.6%) | 0.1 | 3.8 | 0.2 | 0.2 | 4.9 | 17.8 | 34.3 | 43.0 |
| Ours (3.1%) | 0.3 | 8.7 | 0.6 | 0.7 | 9.7 | 19.7 | 37.8 | 43.1 |
| Ours (6.2%) | 0.6 | 20.4 | 2.0 | 2.2 | 24.2 | 21.7 | 36.9 | **43.9** |
| Ours (12.5%) | 1.1 | 34.7 | 5.0 | 6.2 | 45.2 | 22.8 | 37.6 | 42.0 |
| Ours (25.0%) | 2.2 | 47.9 | 10.4 | 12.4 | 64.1 | 23.1 | 39.7 | 41.6 |
| Ours (50.0%) | 4.4 | 58.7 | 16.6 | 19.7 | 76.0 | 23.4 | 39.9 | 41.0 |
| Ours (100.0%) | 8.9 | **66.4** | **22.6** | **27.1** | **82.6** | **24.1** | **40.9** | 41.5 |

# I  EXTENSION TO ARBITRARY VISUAL DOMAINS COVERED BY KNOWLEDGE GRAPHS

One current limitation of this work is its domain restriction to the domain of animals and plants. To demonstrate that our approach is applicable more generally, we describe our current efforts of building a dataset covering the relevant aspects of the entire visual world.

Recall that the LivingThings dataset is built on Wikidata entities related to the root entities "animal" and "plant" as well as the WordNet entities related to the "living thing" node. To build our world dataset, we analyze which entities exist in Wikidata that are not animals or plants (e.g., coffee milk). Then, we find a fitting root entity (e.g., food) and add it to our list of root entities. Next, we find entities that are not animals, plants, or food, and repeat the process. This way, we define 45 root entities. We consider 21 of them as relevant to the visual world and harvest all their related entities, the result of which can be seen in Supp. Tab. 13. Additionally, we show examples of the resulting entities in Supp. Tab. 14 and the skipped root entities in Supp. Tab. 15. Note that entities can be related to multiple root entities, and we keep an entity as long as it is related to at least one of the 21 visual root entities.

We also consider the WordNet knowledge graph and select all "physical objects" that are not a "living thing". However, these entities are mainly professions (e.g., radiobiologist), other terms that characterize humans in some way (e.g., german) or specific people (e.g., Nelson Mandela). We therefore decide to not use WordNet for building our world dataset.

In total, we find around 91k entities with 236k unique terms. Assuming that we obtain around 71 unique images per search (8.9M unique images resulting from 125k search queries as described in Tab. 3) and search for each term, this extends our dataset by around 17M unique images. Additionally, we will generate attributes as described in Sec. 3.2 and download another 17M images for entity-attribute queries. Together with the existing 9M images of animals and plants, this would set our final dataset size at 43M images. This is still considerably smaller than billion-scale CLIP datasets, but with significantly higher quality, allowing for efficient training of world models.

## I.1 QUERYING GENERIC ENTITIES WITH SPARQL

The SPARQL query used to harvest all entities for a root entity is displayed in Appendix I.1. It returns a list of entities from a specified target domain as defined by a root entity. This root entity can be determined manually by searching for appropriate entities on the Wikidata website. For example, if we want to build a model specifically for cars, we set the root entity to motor car (Q1420), as done in Tab. 16.

This query uses the fact that entities within Wikidata are consistently modeled as "instances" and "classes/types". Instances are specific entities of some type or class, e.g., "Barack Obama" is an instance of a "human". On the other hand, classes or types are used to represent sets of entities having something in common and can have sub- or superclasses themselves. For example, the class "human" is a subclass of "mammal", which is a subclass of "animal". With SPARQL we can extract all subclasses and instances under one or more specified root entities, even across multiple hierarchy levels.

```
PREFIX wdt: <http://www.wikidata.org/prop/direct/>
PREFIX wd: <http://www.wikidata.org/entity/>
PREFIX rdfs: <http://www.w3.org/2000/01/rdf-schema#>
PREFIX schema: <http://schema.org/>
PREFIX wikibase: <http://wikiba.se/ontology#>
PREFIX skos: <http://www.w3.org/2004/02/skos/core#>
SELECT DISTINCT
  ?ent
  ?label
  ?desc
  ?links
  (GROUP_CONCAT(DISTINCT ?alias; SEPARATOR=";;;") AS ?aliases)
WHERE {
  VALUES ?typ { wd:Q1420 }
  ?ent wdt:P279* ?typ .
  ?ent rdfs:label ?label .
  FILTER(LANG(?label) = "en")
  ?ent ^schema:about/wikibase:sitelinks ?links .
  FILTER(?links >= 5)
  OPTIONAL {
    ?ent schema:description ?desc .
    FILTER(LANG(?desc) = "en")
  }
  OPTIONAL {
    ?ent skos:altLabel ?alias .
    FILTER(LANG(?alias) = "en")
  }
}
GROUP BY ?ent ?label ?desc ?links
ORDER BY DESC(?links)
```

Figure 11: Generic SPARQL query for extracting entities from Wikidata that are related to a given set of root entities. The root entities should be manually set within the *VALUES ?typ { ... }* clause, here it is just the motor car entity wd:Q1420. A minimum number of sitelinks can also be specified to filter out unpopular entities, here it is set to 5.

Table 13: The root entities for building a dataset to describe the visual world. The "aliases" column refers to the set of all synonyms collected from the Wikidata entities.

| Root entity | Description | Examples | Entities | Aliases |
|---|---|---|---|---|
| product | Anything that can be offered to a market | banh mi, navigation system, PlayStation 2 | 63,676 | 144,715 |
| substance | Any composed matter whose origin is either biological, chemical, or mineral | solid lubricant, Chinese tea, eye cups | 34,259 | 111,383 |
| physical tool | Physical item that can be used to achieve a goal | Patient lift, police transport, instant camera | 32,727 | 71,227 |
| animal | Kingdom of multicellular eukaryotic organisms | saw-scaled viper, Sporathraupis cyanocephala, Rufous mouse-eared bat | 28,000 | 76,408 |
| plant | Living thing in the kingdom of photosynthetic eukaryotes | Whitebark Pine, Eucalyptus coccifera, wig knapweed | 28,000 | 55,925 |
| material | Substance that can occur in different amounts, all with some similar [mixture of some] characteristics, and with which objects can be made | dietary proteins, stone slab tomb, safflower oil | 18,021 | 40,822 |
| vehicle | Mobile machine used for transport, whether it has an engine or not, including wheeled and tracked vehicles, air-, water-, and space-craft | shipwrecks (objects), Evergreen A-class container ship, VTOL aircraft | 17,015 | 37,849 |
| geographical feature | Components of planets that can be geographically located | hydrothermal Vents, grooves, street lamp | 8,683 | 19,030 |
| food | Any substance consumed to provide nutritional support for the body | coffee milk, tikka, Friesian Clove | 8,464 | 15,332 |
| architectural structure | Human-designed and -made structure | rock temples, summerhouse, house of worship | 4,507 | 10,354 |
| anatomical structure | Entity with a single connected inherent 3d shape that's created by coordinated expression of the organism's own dna | bronchi, maxillary wisdom tooth, turtle shell | 4,394 | 9,999 |
| facility | Place, equipment, or service to support a specific function | public toilet, automobile servicing shop, industrial park | 2,767 | 6,740 |
| physical activity | Human physical activity consisting of voluntary bodily movement by skeletal muscles | American rules football, archery, water-skiing | 2,228 | 4,422 |
| clothing | Covering worn on the body | blucher shoe, G-suit, one-piece swimsuit | 1,929 | 4,313 |
| building | Structure, typically with a roof and walls, standing more or less permanently in one place | shoestore, family restaurant, factory outlet | 1,655 | 3,964 |
| musical instrument | Device created or adapted to make musical sounds | electroencephalophone, Chinese flutes, oboe | 1,450 | 3,493 |
| organ | Collection of tissues with similar functions | nasal bone, cranial nerves, ulnar collateral ligament of elbow | 1,155 | 2,450 |
| furniture | Movable objects used to equip households, offices, or shops for purposes such as storage, seating, sleeping | faldstool, airline seat, bicycle parking rack | 388 | 933 |
| body of water | Any significant accumulation of water, generally on a planet's surface | dammed lake, deep-sea hydrothermal vent, marshland | 379 | 792 |
| weather | State of the atmosphere | cold snap, tropical cyclone, sea of fog | 151 | 304 |
| precipitation | Liquid or solid water that falls to the ground | hail, thunderstorm, snowfall | 43 | 72 |
| Total | Before deduplication | | 259,891 | 620,527 |
| Total | After deduplication | | 146,985 | 368,062 |
| New | After deduplication, without animals and plants | | 90,985 | 235,795 |

Table 14: Examples of world entities and accompanying additional information as extracted from the Wikidata knowledge graph. The concrete graph query can be found in Appendix I.1. Name, description and aliases are used as text labels during training. The number of sitelinks are considered a proxy for an entity's popularity. Name and aliases are used as search queries during search.

| Ident. | Name | Description | Sitelinks | Aliases |
|--------|------|-------------|-----------|---------|
| Q3966 | computer hardware | physical components of a computer | 124 | computer component / hardware / computer accessory / PC part / device / computer part / PC component / PC accessory / PC hardware |
| Q81881 | fork | utensil to spear food | 109 | forks |
| Q47616 | incandescent light bulb | electric light using a wire filament heated by a current passing through it, until it glows | 102 | Incandescent Light Bulbs / incandescent light globe / electric lamp / incandescent lamp / incandescent light / light bulb |
| Q5830 | Airbus A380 | wide-body, double-deck, four-engine aircraft, currently the largest passenger aircraft in the world | 100 | Airbus Jumbo Jet / A380 Jumbo Jet / A380 |

Table 15: We consider these root entities either non-visual, irrelevant, or too specific and do not select related entities when building our visual world dataset.

| Root entity | Description |
| --- | --- |
| abstract entity | entity that does not have a physical existence, including abstract objects and properties |
| astronomical object | physical body of astronomically-significant size, mass, or role, naturally occurring in a universe |
| city | large human settlement |
| concept | semantic unit understood in different ways, e.g. as mental representation, ability or abstract object (philosophy) |
| continent | large landmass identified by convention |
| country | distinct territorial body or political entity |
| historical event | particular incident in history that brings about a historical change |
| history | past events and their tracks or records |
| imaginary character | character known only from narrations (fictional or in a factual manner) without a proof of existence; includes fictional, mythical, legendary or religious characters and similar |
| language | particular system of communication, often named for the region or peoples that use it |
| language | structured system of communication |
| medical procedure | process of medicine done to heal; course of action intended to achieve a result in the delivery of healthcare |
| organization | social entity established to meet needs or pursue goals |
| planet | celestial body directly orbiting a star or stellar remnant |
| religion | social-cultural system |
| representation | entity or process that portrays something else, usually in a simplified or approximated manner |
| role | social role with a set of powers and responsibilities within an organization |
| science | systematic endeavor that builds and organizes knowledge, and the set of knowledge produced by this system |
| social system | patterned series of interrelationships existing between individuals, groups, and institutions |
| speciality | field limited to a specific area of knowledge; specialization in an occupation or branch of learning; a specific use |
| star | astronomical object consisting of a luminous spheroid of plasma held together by its own gravity |
| temporal entity | thing that can be contained within a period of time, or change in state (e.g. events, periods, acts) |
| work of art | aesthetic item or artistic creation; object whose value is its beauty only, not practical usefulness |
| written work | any work expressed in writing, such as inscriptions, manuscripts, documents or maps |

Table 16: Car entities and accompanying additional information as extracted from the Wikidata knowledge graph. Showing the first 10 out of 3,549 entities.

| Identifier | Name | Description | Sitelinks | Aliases |
|---|---|---|---|---|
| Q1420 | motor car | motorized road vehicle designed to carry one to eight people rather than primarily goods | 237 | car / automobile / autocar / auto / automobiles / motor vehicle / motor cars / cars / motorcar |
| Q39495 | tractor | engineering vehicle specifically designed to deliver a high tractive effort | 118 | Tractors |
| Q193692 | electric car | automobile propelled by an electric motor using energy stored in rechargeable batteries | 83 | all-electric car / battery-electric car / electric automobile / electrically-powered automobile |
| Q30113 | Jeep | brand of American cars | 65 | |
| Q172610 | Bugatti Veryon | hypersonic car | 63 | Bugatti Veyron EB 16.4 / Bugatti Veyron 16.4 |
| Q182323 | Ford Model T | American car (1908-1927) | 61 | T-Model Ford / Tin Lizzie / Model T Ford / T |
| Q152946 | Volkswagen Beetle | Volkswagen compact car selling over 20 million during its production run from 1936 to 2013 | 59 | Volkswagen Bug / Volkswagen Type 1 / VW Beetle / VW Bug |
| Q243543 | Toyota Corolla | automobile model produced by Toyota | 57 | |
| Q55989 | cabriolet | two-seater or 2 + 2 automobile with a removable roof | 57 | drophead coupé |
| Q188475 | limousine | luxury sedan or saloon car generally driven by a chauffeur | 56 | limo |

