# OpenReview forum: "Leveraging Knowledge Graphs to harvest a high-quality dataset for efficient CLIP model training"
_ICLR.cc/2025/Conference — Submitted to ICLR 2025_

### Official Review · Reviewer_ELED · 2024-11-01

**Soundness:** 3
**Presentation:** 3
**Contribution:** 2
**Rating:** 5
**Confidence:** 3

**Summary:**

This paper proposes a new dataset, named LiveThings, with 8.9M images and 12.2M text scale, leveraging the knowledge graph, Large Language Model, and image search engines. Specifically, the knowledge graphs (e.g., WordNet, Wikidata) output the entity classes and natural types, following the generation of attributes via LLMs. Following the image searching and filtering process, the author proposes a high-quality datasets for animal and plant, which is for biological domain.

**Strengths:**

1. The author proposes a new dataset, LiveThings, to efficiently train the CLIP-based model, which could be leveraged for future works involving foundation model construction.
2. Through extensive experiments, the paper demonstrates the superiority of utilizing automatically generated datasets, especially in efficiency.
3. The paper is well-written, allowing readers to understand easily.

**Weaknesses:**

1. It seems necessary to consider the additional baseline [1]. The proposed dataset, which focuses on biological information containing animal and plant entities, shares similarities in size with another dataset called TreeOfLife [1] that also contains biological data and spans 10M sizes. Therefore, to better demonstrate the advantage of using a knowledge graph in developing their dataset, the authors should perform a comparative analysis with a model that has been trained on the TreeOfLife dataset.

Since it was trained in the biological domain, such as animals and plants, with attribute information, it is natural that it performs well in attribute classification and species classification while showing low performance on other domains (e.g., other in ImageNet). Therefore, it seems necessary to compare it with models trained in the biology domain using a dataset of a similar scale (i.e., TreeOfLife) to validate the effectiveness of leveraging knowledge graphs.

2. There is no experiment for other domains to support the claim that this approach can be used for any domain covered by the given knowledge graph. In the whole paper, only the biological domain is covered, but it is crucial that this approach indeed can cover other domains, such as urban and molecular. The reason for this doubt is that the entities in the knowledge graph may not be diverse in other domains, or it might not be effective in extracting natural types. Therefore, it seems necessary to conduct experiments to prove this.

[1] BioCLIP: A Vision Foundation Model for the Tree of Life. Stevens et al. CVPR'24

**Questions:**

The goal of this paper is to create high-quality data using the knowledge graph, but it seems to be utilized only for extracting entity classes or natural types. To fully leverage the knowledge graph, it appears necessary to use the structural information about the entities within it. Extracting entity-related attributes from the knowledge graph is more natural than using an LLM.
Regarding it, what if the attributes are extracted based on the structural information within the knowledge graph instead of using LLMs? I think it is more natural to fully utilize the knowledge graph's information.

---

> ### Author Response · Authors · 2024-11-20
>
> **W1:** Thanks for pointing out this related work. We ran BioCLIP on all our benchmarks, and ran our model on the Rare Species benchmark proposed by the BioCLIP authors. The results are given in the new appendix chapter "Comparison to BioCLIP". In summary, our model outperforms BioCLIP on all benchmarks except the datasets on which BioCLIP was directly trained on. Our model also works better than BioCLIP on classifying rare species unseen during training. Therefore, we believe that our approach on training on a mix of entities, descriptions, attribute queries, and alt texts leads to a more general model than training only on the entity names.
>
> Note that we will move BioCLIP and the Rare Species dataset from the appendix to the main paper until the end of the discussion period, but have postponed this in the interest of responding to the reviews earlier.
>
> **W2:** To show that our approach can cover other domains, we added a new appendix chapter "Extension to arbitrary visual domains covered by knowledge graphs" on how we extend our dataset to a much larger visual domain. We show the results in table 15, where we obtain 91k new entities from 21 root entites with 236k alias terms. With this diverse data, we can use our approach to scale our dataset to an expected total of 43M images.
>
> For obtaining the natural types of the entities, we can use the root entity as a baseline ("food" as the natural type of "coffee milk") and use our approach from chapter "query building": We select the parent hierarchy of the entity from the knowledge graph, and use an LLM to decide based on entity name, description, and parents, which parent fits best as the natural type. Unless the entity is completely unknown to the LLM, or the hierarchy in Wikidata is really low quality, which are both unlikely scenarios, this approach will work just as well as it did for the animal and plant domain.
>
> **Q1:** The main problem is that attributes are heterogeneously distributed and sparsely annotated within the Wikidata knowledge graph. Many entities, e.g., the [car tire](http://www.wikidata.org/entity/Q43716070) entity have no attributes annotated at all. Even if attributes exist, there is no obvious way to distinguish visual from non-visual attributes, e.g., the [tram](https://www.wikidata.org/wiki/Q3407658) entity provides potentially useful visual attributes (that it has a door and a driver's cab) but also more abstract non-visual ones (its average carbon footprint during operation, and that it is a facet of urban transport). Finally, attributes are annotated with many types of property relations: has characteristic, has use, has parts, used by, source of, habitat, means of locomotion etc., which adds complexity in automatically extracting attributes.
>
> On the contrary, querying LLMs automatically is straightforward. Since they acquire comprehensive world knowledge during pretraining, we consider them as a viable alternative for mining attributes.
>
> We also would like to note that attributes are often already captured via the entities itself. E.g., the entity [tire](https://www.wikidata.org/wiki/Q169545) has subclasses like [flat tire](https://www.wikidata.org/wiki/Q1779337) or [snow tire](https://www.wikidata.org/wiki/Q115170), which our current approach makes use of.

---

> ### Comment · Reviewer_ELED · 2024-11-26
>
> Thank you for your sincere response.
> However, I still have concerns regarding this work.
> 1. After reviewing the responses to W1 and Q1, I feel that stating that the Knowledge graph was used to obtain high-quality data is somewhat of an overstatement (it is even mentioned in the title). It seems that the Knowledge graph was primarily used for entity extraction. As the author mentioned in W1, the reason for better performance compared to BioCLIP is not due to the Knowledge graph but rather the use of descriptions, queries, and alt text - these are not derived from the Knowledge graph but from the LLM.
> 2. Regarding W2, is there any guarantee that the generated dataset for the new domain is of high quality? While it performed well in the animal and plant domains and showed superior performance, I'm concerned that stating that the labeled dataset for new domains is of high quality since this dataset was derived in the same way might be an exaggeration.
> 3. I share a concern similar to Reviewer rB9j's comment. There is a possibility that the retrieved images might be unaligned, and it seems important to check for this. This is because, in the ablation study in Table 6, when only Google Search is used, the performance is poor, and I suspect this may be due to the retrieval of unaligned images from Google Search. Therefore, I think this issue is quite important and should be addressed.

---

> ### Author Response · Authors · 2024-11-27
>
> **1:** To clarify, the entity description, name and synonyms are given by the knowledge graph (KG). The alt text is given by the HTML "alt" tag of the image during downloading, same as with other CLIP datasets. Natural types (e.g., "animal" as natural type for "dog") are proposed by the KG and selected by the LLM. Attributes are generated by the LLM, given the entity name, synonyms and definition from the KG as description. Note that in all our LLM prompts we give the KG information as context, so without the KG we could not prompt the LLM as we do now.
>
> During training, we sample text labels as described in figure 2: We train on entity descriptions and synonyms (both given by KG), attribute-and-noun (noun synonym given by KG, attribute given by LLM), and the noun-attribute query generated by LLM. So, we do use the KG labels during training as part of a mix of KG labels, LLM-generated text, and alt text.
>
> Additionally we would like to mention that entity extraction itself is already useful. Related works like MetaCLIP extract queries by selecting e.g. the most common words and word pairs from Wikipedia, which also includes stop words and instances of specific people. KG entities on the other hand always describe entities, they include a definition and synonyms, and they include a hierarchy which we can use to extract the natural type. We also specifically avoid selecting specific instances of people or animals, and only select the class (e.g., select dog but avoid a specific entity like the dog "Lassie").
>
> In summary, the KG is the main information source in our dataset building process, and provides labels for training.
>
> **2:** To estimate the quality of the dataset on the new domain we can consider the potential sources of problems, i.e, why the dataset in the new domain could be of worse quality, than in the animal and plant domain.
>
> We verified that the Knowledge Graph is big enough to extend the domain in our new appendix chapter. Upon qualitative inspection the entities also are of similar quality, e.g., [steam tractor](https://www.wikidata.org/wiki/Q382477), [banoffee pie](https://www.wikidata.org/wiki/Q2748774), and [baroque violin](https://www.wikidata.org/wiki/Q808578) all show proper description, name, and the root entities are also correct (vehicle, food, and musical instrument respectively).
>
> We use LLMs for the same purposes as before: Selecting natural types and generating attributes based on name, synonyms, and description. LLMs are trained on internet-scale data without domain restrictions, so there is no immediate reason as to why LLMs should have less understanding of domains like vehicles or food, than of animals and plants. Additionally we can use more recent LLMs to improve the generation results, so potentially the results will be better than before.
>
> Upon inspection, the Bing image search also works well on the new domain, e.g., [stream tractor](https://www.bing.com/images/search?q=steam+tractor) or [baroque violin](https://www.bing.com/images/search?q=baroque+violin).
>
> Finally, the alt-text quality might vary. However, alt-text quality affects all CLIP models equally, so it would also affect the SOTA Models we compare to. There are works that improve alt-texts e.g. by using LLMs to rewrite them [7], but we consider this orthogonal research, as it could be applied to all CLIP models we test.
>
> To conclude, our approach will most likely create a dataset of similar or better quality on the new domain, as it does on the animals and plants domain.
>
> [7] Li et al., Aria: An Open Multimodal Native Mixture-of-Experts Model, arXiv:2410.05993, 2024.
>
> **3:** We agree that the Google image search is of lower quality than the Bing image search:
> At the end of chapter "Results" in paragraph "Ablation studies" we mention that "the search results of the Google API are significantly worse than the results of the Bing API" which was the main reason we switched to Bing search. This is confirmed by the results in the ablation study in Table 6.
> So our quality check of random samples uncovered the low quality of the Google API results.
> One failure example is "spotted dog", where the Google API produces images of restaurants with this name, other animals, and dogs without spots, while the Bing API properly produces dogs with a spotted fur pattern.
>
> Additionally, we would like to refer to our response to Reviewer rB9J's concerns.
>
> We uploaded a random sample of 20k datapoints and a utility to view the data in the supplementary,
> and invite the reviewers to check the quality of the data.
> We find that for automatically collected internet data, the quality is high, especially for the Bing API results.

---

### Official Review · Reviewer_nJby · 2024-11-02

**Soundness:** 3
**Presentation:** 3
**Contribution:** 3
**Rating:** 6
**Confidence:** 3

**Summary:**

This paper proposes to harvest small-scale but high-quality data for CLIP training leveraging existing knowledge graphs. For a proof of concept, the paper focuses on visible living things including animals and plants. By extracting entities from knowledge graph, generating attributes and building queries using LLM, and collecting images using Google API, the authors created a dataset named LivingThings with 8.9M images and 12.2M texts. With the collected dataset, the authors trained a CLIP ViT-Base model from scratch and achieve comparable or better performance over large-scale trained CLIP variants.

**Strengths:**

1. The paper is well organised with clear motivation. The overall flow is easy to follow.

2. The work contributes a high-quality dataset named LivingThings covering animal and plant entities, which I believe would benefit the community.

3. This work is quite resource-efficient, training a CLIP-style model for visual recognition using only 8 RTX 2080 Ti GPUs. And each round of training only costs ~30 hours.

**Weaknesses:**

1. The model studied in this work is ViT-Base, a small variant of the CLIP family. Therefore, the scalability of model size using the LivingThings dataset might need further verification.

2. The method relies on the prior knowledge of the root entity (e.g., plant and animal) when collecting the dataset. Therefore, the proposed paradigm seems only suitable for visual recognition in a single or limited number of domains.

**Questions:**

1. LivingThings has a total of 8M images. I am curious about the model performance of using even fewer images, e.g., 100K, or 1M.

2. How would the proposed paradigm be extended to build a CLIP-level model that covers unlimited domains? And how many images might be needed to match the performance of standard CLIP models? Just some analyses would suffice.

---

> ### Author Response · Authors · 2024-11-20
>
> **W1:** Training a ViT-Large even on our smaller scale dataset would take several weeks on our hardware, which would leave us without ability to do ablation studies or other analyses. Additionally, given the work "Training compute-optimal language models" [5], we would assume that when scaling up the model, we would also need to scale up the data. This has been shown in "Scaling (Down) CLIP" [6], see figure 8a on page 7: The break-even point between ViT-Base and ViT-Large model for a standard CLIP dataset is at around 400M image-text pairs. So at the current dataset scale and, considering we want to train an efficient model on small hardware, we regard ViT-Base a good choice.
>
> In our main paper (table 7) we compare various smaller models and different patch sizes. The results show that our dataset scales well to the most costly model ViT-B-16.
>
> **W2:** We added a new appendix chapter "Extension to arbitrary visual domains covered by knowledge graphs" on how we extend our dataset to a large visual domain similar to CLIP. Indeed, we manually define all our root entities, but we do not need expert knowledge of these domains -- only the expertise on how to query the knowledge graph itself. In summary, we look at the set of entities that we have not selected yet, find a common root entity and select the entities related to this root entity. Since the total number of root entities needed is quite small, this manual effort is negligible compared to the amount of entities we obtain. The root entities resulting from this procedure can be seen in table 15.
>
> **Q1:** Thanks for the suggestion. We varied the size of the dataset and added the results of this study to the appendix "scaling study".
> The results show that each time we double the dataset size, we get a significant improvement in model performance.
>
> **Q2:** For the extension of the proposed paradigm to build a CLIP-level model, please refer to the answer to weakness 2 above.
>
> As for what number of images might be needed to match CLIP:
>
> Considering that we can already match OpenAI CLIP on the Living subset of ImageNet with only 9M images, and assuming that living things cover 10-20% of all known objects, we expect to get a model covering the same domains as CLIP with 40-80M images, maybe even less because of shared terms and features. The dataset described in appendix chapter "Extension to arbitrary visual domains covered by knowledge graphs" would result in 43M images.
>
> Related to this, Scaling Down CLIP [2] use CLIP to filter the top quality images of their base 3.4B dataset and show in figure 6 and 7 that, depending on the setting, dropping e.g. the lower 50% of data can even improve the model. We believe that with our focused search queries we can search for the right data instead of just taking more or less random image-text pairs from the web, and match OpenAI CLIP with 10-20% of their dataset size.
>
> **References:**
>
> [2] Gadre et al., Datacomp: In search of the next generation of multimodal datasets. NeurIPS, 2023.
>
> [5] Hoffmann et al., Training Compute-Optimal Large Language Models. arXiv:2203.15556, 2022.
>
> [6] Li et al., Scaling (Down) CLIP: A Comprehensive Analysis of Data, Architecture, and Training Strategies. arXiv:2404.08197, 2024.

---

> > ### Comment · Reviewer_nJby · 2024-11-24
> >
> > Thanks for the rebuttal. My questions have been addressed, I will keep my positive score.

---

### Official Review · Reviewer_rB9j · 2024-11-04

**Soundness:** 2
**Presentation:** 3
**Contribution:** 3
**Rating:** 5
**Confidence:** 4

**Summary:**

This paper tries to propose a controlled-generated image-pair dataset, where the query entity is from the Wikidata knowledge graph, and the attribute and search query are generated using LLMs. Then, search engines are used to collect the images. Later, it proves that the 8.9M dataset can train the model with comparable performance of larger dataset on living things categories.

**Strengths:**

1.	The goal is to investigate whether a high-quality small-scale dataset can achieve comparable performance to the CLIP model trained on large data.
2.	Create the datasets that are based on the Wikidata knowledge graph, ensuring wide data coverage over different categories.
3.	The query creation + searching is a good way to construct high-quality dataset.

**Weaknesses:**

1.	This paper collected the LivingThings dataset covering animals and plants, but it is still small scale and the model trained from it cannot compare with SOTA CLIP model. I wonder if anything this paper can do to further improve the SOTA CLIP model by using this datasets?
2.	This dataset only has a local coverage of animals and plants. But I wonder how can this data collection pipeline be further scaled up to the next DataComp or LAION sized dataset. If the author keeps increasing the entity number from wikidata, it might be piled up by including other categories, and finally still result in a very large dataset in billion scale, then you will have to prove it is better than LAION or DataComp
3.	I would expect more detailed experiment analysis and ablation study about the effect of attribute number, number of entities, scaling law, etc.
4.	The quality control of data collection seems to be loose. Seems it cannot guarantee the alignment of text and image is high. Why not use a CLIP to filter out the unaligned image-text pair?

**Questions:**

I think the main concern is that the dataset itself is less in data analysis and experimental ablation. Also, it cannot surpass the SOTA CLIP, and didn't show clearly how it can help boost SOTA CLIP.

---

> ### Author Response · Authors · 2024-11-20
>
> **W1:** To investigate whether we can further improve SOTA CLIP with the existing dataset, we finetune the DataComp-1B CLIP model on our LivingThings dataset and report the results in appendix "Finetuning results", in tables 11 and 12. Notably we get a model that is better on finegrained animal and plant classification than both the pretrained DataComp-1B model and our pretrained LivingThings model. So, we can indeed improve SOTA CLIP models by finetuning them on our high-quality dataset.
>
> However unlike finetuning, pretraining allows full control over model architecture and input data. One of our goals is to enable efficient pretraining of CLIP models on smaller scales. In this limited hardware setting, we believe it is useful to be able to pretrain strong models from scratch, even if they don't beat the biggest models on all benchmarks.
>
> Note that we will add these finetuning results to the main paper until the end of the discussion period, but have postponed this in the interest of responding to the reviews earlier.
>
> **W2:** We added a new appendix chapter "Extension to arbitrary visual domains covered by knowledge graphs" on how our dataset can be extended to a large visual domain similar to CLIP. In table 15 we show the root entities used to harvest additional entities, e.g., physical tool or architectural structure, and the resulting number of entities. With our approach based on knowledge graphs and image search, we will scale our dataset to around 43M images, which is still magnitudes below the billion-scale CLIP datasets. Out of these 43M images, around 20% will be animal and plant data, which we believe is a reasonable amount considering the abundance of animal and plant data on the internet.
>
> The goal of our work is to demonstrate how to create a comparatively small-scale dataset to efficiently train a foundation model for new domains that CLIP does not cover well. Since we have successfully done this for the animals and plants domain, we believe our final dataset will be a strong contender to the bigger datasets, especially when considering resource efficiency.
>
> Alternatively, we could try to scale up further by exchanging image search to caption filtering of raw web data similar to the original CLIP work [1], DataComp [2] or MetaClip [3]. Also, we could use our training model as a filter similar to DataFilteringNetworks [4]. However these approaches would have little value since many of such datasets and models already exists, and we would run into the high compute demands of billion-scale datasets that we explicitly want to avoid.
>
> **References:**
>
> [1] Radford et al., Learning transferable visual models from natural language supervision. PMLR, 2021.
>
> [2] Gadre et al., Datacomp: In search of the next generation of multimodal datasets. NeurIPS, 2023.
>
> [3] Xu et al., Demystifying CLIP data. ICLR, 2024.
>
> [4] Fang et al., Data Filtering Networks. ICLR, 2024

---

> ### Author Response · Authors · 2024-11-20
>
> **W3:** Thanks for the suggestions. To investigate the scaling law, we train our model at various scales of our dataset, and add the resulting figure and table into a new appendix chapter "Scaling study". As expected, the model gets significantly better each time we double the dataset size.
>
> As for numbers of attributes and entities, we show various ablations in table 6: Training without attribute data, training without WordNet entities, training with only search results from one of the two search engines. In all cases, reducing the number of entities, attributes or images leads to a weaker model.
>
> Please let us know if you have other suggestions on how to investigate our dataset building process.
>
> **W4:** We explicitly refrained from using the existing CLIP models: a) We might incorporate its weaknesses. If we use existing CLIP to filter, and it cannot understand some of the data, it might filter it out. We show that pretrained CLIP models are lacking in distinguishing finegrained animals and plants, so potentially they are suboptimal filters for such data. b) When using CLIP for building the dataset, we cannot demonstrate anymore how to build a model from a much smaller dataset, without already having a model.
>
> Additionally, the image search engines already provide strong alignment between image and text.
> We qualitatively analyzed the search results and found that in most cases both the image matches the search query, and the alt-text matches the image, see the randomly picked samples in appendix figures 6 and 7.
> Even complex queries, e.g., [maple tree along a road](https://www.bing.com/images/search?q=maple+tree+along+a+road), give the desired results.
>
> One failure case we observe is that if the attribute-noun combination cannot be found, the search engine returns just the noun, e.g., [short beech tree](https://www.bing.com/images/search?q=short+beech+tree) returns regular beech trees and ignores "short". However, CLIP does also not understand such finegrained differences well enough to work as a better filter: The CLIP ViT-L Similarity between [this picture of a european beech](https://th.bing.com/th/id/OIP.xfMYDPdoiVYvBATVRNm5FwHaE7?w=274&h=182&c=7&r=0&o=5&dpr=2) and the texts "european beech", "short beech", and "a photo of a european beech" are all below the commonly used similarity threshold of 0.3, so CLIP would not be able to even properly filter images for this entity, let alone the entity-attribute combination.

---

> > ### Comment · Reviewer_rB9j · 2024-11-25
> > **Reviewer's Feedback**
> >
> > Thanks for the explanation and I really appreciate it, especially the more experiments on W1 and W3
> > However, part of my concern is still not fully addressed.
> > 1. The construction of the dataset can be generalized to more categories from WikiText and leads to 43M images, however, such an experiment on generalized categories is not done and I am not sure how well it can outperform the OpenCLIP trained on LAION5B, because the searched image is of the limit to the complexity of query. It might have more images to be searched if the query's scene graph is simple. However, many complex scene image is not covered by the proposed data construction pipeline, especially multiple objects, etc. But LAION data might cover those.
> > 2. We can definitely improve the complexity by constructing a more complex query, however, the retrieved image might be more likely to be unaligned with the query. The author still does not provide a valid way to do the quality check but only relies on search engines.

---

> ### Author Response · Authors · 2024-11-27
>
> **1:** To restate our goal, we do not necessarily aim to outperform the Laion5B model trained on billions of images and hundreds of A100 GPUs with our model on all benchmarks. Instead, we aim to create a relatively small dataset and enable efficient training of CLIP foundation models on small-scale hardware. In our existing experiments we show that we can learn a high-quality model about objects and attributes from scratch, which validates our approach. For another validation we would like to refer to the experiments requested by reviewer ELED and performed in chapter "Comparison to BioCLIP", where our model strongly outperforms a model built with biological domain knowledge, even though in our case all domain knowledge stems from the knowledge graph.
>
> Regarding the query construction, it is true that in our pipeline we search for objects and objects with various properties (color, texture, shape, environment, etc.).  We do not explicitly search for complex queries, since the more complex the query, the less likely it will be that we find fitting images in the search engine. This problem is described in the last paragraph of our answer to our point W4 (the search engine will start to ignore parts of the query, if it cannot find results for the entire query). So it is correct that we cannot search for arbitrarily complex scene graphs, since there might not exist images for those.
>
> However importantly, we search for simple queries, but still download all corresponding alt-texts for the image. Recall that works like original CLIP [1], MetaCLIP [3], or Laion5B collect their data by filtering alt-texts either using substring matching with simple queries (1 or 2 words), or by using CLIP similarity. So, the complex scenes that Laion5B covers, come from internet alt-text. Our dataset contains the same kind of alt-texts, so our training dataset contains similarly complex scenes as Laion5B, to an extent. To verify this, the third image in figure 1 shows "a group of horses running in the snow" when searching for "running animal". This is a scene with multiple objects, an activity, and an environment. So it is significantly more complex than the search query "running animal".
>
> **2:** We agree with the point that more complex queries might lead to worse results in the image search engine as described in the answer above, which is why we refrained from increasing the query complexity above object-and-property pairs.
>
> Our quality checks so far consists of using the image search engine, simple heuristics (text should not be empty, image should have a minimum resolution, etc.), and a qualitative manual check on a subset the search results, which verified that the search engine returns aligned pairs.
>
> Works like the original CLIP and MetaCLIP are successfully creating pretraining datasets by only employing filtering alt-texts with queries, and using similar simple heuristics. Especially they do not employ existing VLMs. A modern search engine gives better results than a simple substring matching, so our dataset will be of higher quality than the original CLIP dataset and MetaCLIP, simply by replacing substring matching with image search engines.
>
> Laion5B and others like DataComp [2] use CLIP for filtering, which we also would like to avoid as described in our answer to W4. Similar problems would arise if we use off-the-shelf e.g. object detectors. We do not want to rely on existing models to build our foundation model. Manual human filtering of our entire dataset would contradict our goal of building the dataset automatically using knowledge graphs, and would contradict our limited resource scenario due to high cost (at least 100k USD to filter only the current 9M images).
>
> Finally we would like to note that an automatically collected CLIP pretraining dataset does not need to be of perfect quality.
> Our experiments show that our model learns well given the small amount of images, so we reached a dataset quality high enough to learn a strong model.
>
> We would like to give the reviewers the opportunity to look at our dataset themselves, so we uploaded a random sample of 20k datapoints and a utility to view the data in the supplementary. We invite the reviewers to check the quality of the data.
> We find that for automatically collected internet data, the quality is high, especially for the Bing API results.

---

### Author Response · Authors · 2024-11-20

We thank the reviewers for their insightful comments and questions and for taking time to review our work. We have uploaded a new submission and added new content to the appendix beginning on page 24. Notably, we compared our approach to BioCLIP and compared all models on the Rare Species dataset, created a study on dataset scale, and described how we can extend our data domain from animal and plant data to the entire visual world. We also compare pretraining and finetuning on our dataset. All new content is indicated in blue, for new sections we mark the section title, and for new figures and tables we mark the caption.

In the interest of responding to the reviews now, and due to the page limit, we have postponed integrating the new content into the main paper and have added everything to the appendix. We will integrate parts of the appendix content into the main paper until the end of the discussion period, notably we will update the main paper with the results of BioCLIP, finetuning, and the Rare Species dataset.

---

> ### Author Response · Authors · 2024-11-27
>
> Again, we thank the reviewers for the insightful discussion. To enable the reviewers to look at the LivingThings dataset, we uploaded a random sample of 20k datapoints and utility code to view the data in the supplementary.

---

> ### Author Response · Authors · 2024-11-28
>
> We have integrated the new content created during the discussion into the main paper in our latest revision. To clarify, we did not add any new content in this revision, but moved content from the supplementary to the main paper.

---

### Meta-Review · Area_Chair_neSZ · 2024-12-21

**Metareview:**

This work proposed to leverage knowledge graph as the guidance to collect high-quality data for CLIP model training, particularly focused on animal and plant categories. The authors start with a knowledge graph and further extend the query by adding attributes and types using LLMs. The resulted dataset called LivingThings consists of 8.9M images and 12.2M texts. Afterwards, the authors trained a CLIP model from scrach on the dataset and evaluate it on various benchmarks including ImageNet, iNaturalist and CUB for image classification. The trained model showed superior performance on these benchmarks and even outperforms those trained with much larger dataset and more parameters.

The main strength of this work is introducing a high-quality dataset LivingThings which is the first ever vision-language dataset that are specifically designed for animals and plants. Based on this dataset, the authors pretrained a CLIP model from scratch and demonstrate good performance acorss various benchmarks.

As pointed out by the reviewers, however, the proposed method of constructing image-text data for CLIP training is not necessarily generalized to new domains. Despite some efforts to prove this during the rebuttal, it still cannot resolve fully the reviewers' concern. In additiona, the diversity of the collected data is also concerned as all the queries are generated based on the entities in the knowledge graph. At last, the ACs also think this work does not show the generality of the proposed method if the goal is to leverage knowldege graph to harvest a high-quality dataset for CLIP training, regardless of the domain of interests.

After the rebuttal session, this work got 5,5,6 from three reviewers. Based on the reviews and discussions, the ACs think this work still has some obvious flaws, as thus suggest it is not ready for this venue.

**Additional Comments On Reviewer Discussion:**

There are some discussions between the reviewers and authors during the rebuttal session. Two out of three reviewers still have some concerns unresolved after the rebuttal. The ACs agree with the reviewers that this work makes contributions to the community but is not sufficient to appear in this venue given its current stage. The ACs highligh recommend the authors integrate the additional experiments to the main submission, conduct more rigrious analysis and comparisons on the collected data, and more importantly showcase the uniqueness of the proposed method and dataset.

---

### Decision · Program_Chairs · 2025-01-22

Reject